# HieRD: Hierarchical Relational Distillation for Vision-Language Embedding Models

Vinh Le [1] [*]   Nguyen Hong Dang [1] [*]   Tu Vu [2]   Linh Ngo Van [1]   Duc Anh Nguyen [1]   Trung Le [3]

## Abstract

Knowledge distillation is crucial for compressing large Vision–Language Models (VLMs) into efficient architectures. While prior VLM research has primarily focused on reasoning tasks like visual question answering, multimodal embedding learning, a key component for large-scale retrieval, has received comparatively less attention. Existing distillation methods typically align static global representations, overlooking hierarchical feature structure and fine-grained cross-modal interactions. This leads to a structural gap where student models fail to inherit object-level semantics and spatial relationships from teachers. To address this limitation, we propose **HieRD**, a Hierarchical Representation Distillation framework that preserves hierarchical structure within and across modalities throughout the distillation process by leveraging clustered visual tokens and multi-granular alignment with phrase-level text. Experimental results on multimodal embedding and downstream tasks show that HieRD consistently outperforms strong baselines, reflecting the effectiveness of its fine-grained semantic and spatial modeling, while enabling compact and efficient embedding models.

## 1. Introduction

Recent advances in vision–language foundation models have substantially expanded the capabilities of multimodal AI, enabling strong performance on generative and reasoning-centric tasks such as visual question answering, image captioning, and multimodal instruction following across diverse benchmarks (Li et al., 2025; Yang et al., 2025; Bai et al., 2023; Wang et al., 2024; Bai et al., 2025). Despite these successes, the large scale and computational demands of such models severely limit their deployment in real-time and resource-constrained settings. In contrast, multimodal embedding models, which encode images and text into a shared semantic space for efficient retrieval, remain comparatively underexplored despite growing practical demand (Zhang et al., 2025a; Li et al., 2026; Zhang et al., 2025b). These models are critical for applications such as cross-modal search, semantic indexing, and recommendation systems, where efficiency and scalability are paramount. While recent efforts such as VLM2Vec (Jiang et al., 2025) have begun to establish benchmarks for vision–language embeddings, state-of-the-art performance still relies on large, resource-intensive architectures that are impractical for deployment.

Knowledge distillation (KD) (Hinton et al., 2015) provides an effective framework for compressing large language and vision-language models into efficient student architectures. However, most existing KD methods assume architectural similarity and shared tokenization between teacher and student (Hinton et al., 2015; Gu et al., 2024; Feng et al., 2025). Recent efforts relax these assumptions through cross-architecture alignment mechanisms such as optimal transport (Boizard et al., 2025) or fine-grained token correspondence modeling (Wan et al., 2024). More broadly, KD for language-only LLMs and LLM-based embedding models has received substantial attention, with recent studies exploring cross-tokenizer alignment (Vu et al., 2026a), cross-architecture and optimal-transport-based transfer (Hoang et al., 2026), preference distillation (Nguyen et al., 2026), token-level self-improvement (Le et al., 2025), span- or trajectory-level alignment (Dao et al., 2026b; Chi et al., 2026; Dao et al., 2026a), and embedding-model representation alignment (Truong et al., 2025; Vu et al., 2026b; Huy et al., 2026). These studies highlight the growing interest in KD for LLMs and embedding models. However, distillation for compact vision–language embedding models remains comparatively less explored, especially when the goal is to preserve hierarchical cross-modal relational structure

In vision-language models, prior distillation approaches

---

[*]Equal contribution   [1]Hanoi University of Science and Technology, Hanoi, Vietnam [2]The Australian National University, Canberra, Australia [3]Monash University, Clayton, VIC 3800, Australia. Correspondence to: Duc Anh Nguyen <anhnd@soict.hust.edu.vn>, Trung Le <trunglm@monash.edu>.

*Proceedings of the 43rd International Conference on Machine Learning*, Seoul, South Korea. PMLR 306, 2026. Copyright 2026 by the author(s).

have explored cross-modal knowledge transfer, but their primary focus remains on enhancing generative or reasoning capabilities rather than learning compact multimodal embeddings. For example, Align-KD (Feng et al., 2025) distills cross-modal alignment knowledge by guiding the student to mimic teacher attention distributions and cross-modal projections to improve downstream task performance. In contrast, EM-KD (Feng et al., 2026) addresses unbalanced vision tokens between teacher and student models by matching semantic and affinity relationships; however, it lacks an explicit and interpretable mechanism for identifying which visual tokens correspond to the same underlying semantic entity across models.

These limitations are further compounded by the fact that most existing frameworks implicitly treat all tokens uniformly, ignoring the hierarchical and relational structure inherent in both language and vision. Linguistic theory has long established that natural language exhibits hierarchical organization, with words forming nested phrases and higher-level semantic constructs (Crain & Nakayama, 1987; Everaert et al., 2015). Similarly, interpretability studies of transformer models suggest that textual and visual representations progressively organize into semantically coherent clusters across layers (Jawahar et al., 2019), where lower layers capture fine-grained lexical or patch-level features and higher layers synthesize these into abstract and object-level concepts (Montavon et al., 2018). Ignoring this intrinsic hierarchy leads to student models that mimic isolated teacher behaviors without capturing the representational trajectories essential for semantic understanding and generalization (Kovalev & Tikhomirov, 2025).

To address these gaps, we propose **HieRD**, a **Hie**rarchical **R**epresentation **D**istillation framework tailored for vision–language embedding models. **HieRD** explicitly captures hierarchical cross-modal structure by grouping visual tokens into semantically coherent clusters that approximate object-level entities and aligning them with phrase-level textual spans. Rather than enforcing flat token-wise matching, our framework introduces multi-granular relational alignment objectives that preserve both intra-modality structure and cross-modal correspondence at multiple levels of abstraction. By abstracting token representations into cluster-level semantic units, HieRD naturally accommodates architectural differences between teacher and student models, enabling meaningful alignment even when token counts or layer structures differ.

We further provide theoretical motivation showing that transformer-based vision encoders naturally develop cluster-like representations as depth increases, which can serve as stable and interpretable anchors for hierarchical distillation. In summary, our contributions are as follows:

- We identify key limitations in existing vision–language distillation methods, including inadequate modeling of hierarchical structure and ambiguous token correspondence across heterogeneous architectures.

- We introduce **HieRD**, a hierarchical relational distillation framework that aligns clusters of visual tokens with phrase-level textual spans to preserve fine-grained cross-modal semantics.

- We provide theoretical motivation for exploiting cluster structures in transformer models as effective anchors for hierarchical distillation.

- Extensive experiments demonstrate that HieRD consistently outperforms strong distillation baselines on diverse multimodal embedding tasks, enabling compact and efficient models with high-quality semantic representations.

## 2. Background

### 2.1. Knowledge Distillation

Originally formalized by (Hinton et al., 2015), Knowledge Distillation is a widely used framework for model compression, aiming to transfer the predictive behavior of a high-capacity teacher to a more efficient student model (Wilf et al., 2025; Gu et al., 2024; Wan et al., 2024; Zhang et al., 2024). In multimodal embedding learning, KD is commonly adopted to construct a shared semantic space that aligns visual and textual representations, serving retrieval and similarity-based tasks.

Given a mini-batch of multimodal input which jointly constructed from visual and textual content, a contrastive loss $\mathcal{L}_{con}$ (Jiang et al., 2025) encourages representations of corresponding sentences to be more similar than those of non-corresponding ones, thereby promoting discriminative semantic separation in the embedding space.

However, distillation methods that focus solely on matching final embeddings often fail to capture the relational structure of the teacher's representation space. To mitigate this issue, Park et al. proposed Relational Knowledge Distillation (RKD), which replaces point-wise alignment with relational supervision over mini-batches. By preserving relative distances and angular relationships induced by the teacher, RKD enables the student to better maintain the global geometry of the embedding space.

Accordingly, we define the base training objective as

$$\mathcal{L}_{base} = \mathcal{L}_{con} + \alpha \mathcal{L}_{RKD}, \qquad (1)$$

where $\alpha$ balances the contribution of relational supervision. The detailed formulation of RKD is provided in Appendix B.

## 2.2. Hierarchical Representations in Language and Vision

Natural language is inherently hierarchical, organizing discrete lexical units into nested syntactic and semantic structures. Linguistic theories and empirical studies have long demonstrated that phrases and clauses serve as fundamental units of meaning, capturing semantic compositionality beyond individual words (Socher et al., 2011; 2013). In recursive and compositional models, meaning is constructed in a bottom-up manner, progressively merging lower-level units into higher-level abstractions (Tai et al., 2015).

Recent analyses of Transformer-based architectures reveal that such hierarchical organization also emerges implicitly in deep neural networks. In language models, lower layers tend to encode surface-level lexical or syntactic information, while higher layers capture increasingly abstract semantic representations (Tenney et al., 2019; Rogers et al., 2020; Clark et al., 2019). This layer-wise specialization is further amplified in large language models, where early layers emphasize lexical and factual retrieval and deeper layers support complex reasoning and abstraction (Wang & Choi, 2023; Jin et al., 2025). Collectively, these findings suggest that representation evolution across Transformer layers parallels the construction of a linguistic parse tree, progressing from fine-grained tokens to semantically coherent structures.

In the visual domain, Vision Transformers process images as sets of local patch tokens. Although these tokens are initially spatially localized, studies in representation analysis and interpretability indicate that tokens corresponding to the same object or semantic region tend to become increasingly correlated across layers, implicitly forming object-centric groupings (Caron et al., 2021). This emergent behavior highlights that both language and vision representations exhibit hierarchical and compositional structures (Naseer et al., 2021), which are not adequately captured by flat, token-level alignment strategies.

## 2.3. Clustering as Structural Abstraction in Representation Spaces

Clustering provides a natural mechanism for abstracting fine-grained representations into higher-level semantic units. By grouping elements that are closely related in the feature space, clustering enables the transition from local representations to more structured and interpretable abstractions.

Among various clustering techniques, density-based methods such as DBSCAN (Ester et al., 1996) are particularly well-suited for high-dimensional and irregular data distributions. Unlike partition-based algorithms, DBSCAN does not require specifying the number of clusters in advance and can adapt to varying cluster densities, making it robust to noise and outliers.

## 3. Methodology

### 3.1. Theoretical Analysis of Visual Token Clustering in Self-Attention

**Problem Setup and Notation.** We consider a VLM composed of a vision encoder followed by a multimodal projector. The vision encoder maps an input image into a sequence of visual patch tokens updated through stacked Transformer layers with unimodal self-attention, prior to any projection into a shared multimodal embedding space or cross-modal interaction with text.

Let $Z_t = \{z_1^t, \ldots, z_N^t\} \subset \mathbb{R}^D$ denote the set of visual token embeddings at layer $t$, where $N$ is the number of image patches and $D$ is the hidden dimension. To analyze the geometric behavior of self-attention, we assume that the value matrix satisfies $\|W_V\|_2 \leq 1$. Under this assumption, each token at layer $t + 1$ is updated as a convex combination of token embeddings from the previous layer, $z_j^{t+1} = \sum_{k=1}^N a_{k,j}^t W_V z_k^t$, where $a_{k,j}^t$ denotes the attention weight from token $k$ to token $j$ at layer $t$.

**Motivation and Analytical Objective.** Self-attention aggregates token representations through weighted averaging, encouraging tokens associated with the same object or coherent semantic region to move closer in the embedding space, while interactions across distinct regions are typically weaker. This observation raises a fundamental question: *does unimodal self-attention within the vision encoder induce implicit clustering among visual tokens, and how does this behavior evolve across layers?*

To formalize this intuition, we study the contraction properties of self-attention by analyzing how the diameter of the visual token set and intra-cluster distances evolve across layers. We define the diameter of a set of vectors $A \subset \mathbb{R}^D$ as

$$\mathbf{d}(A) = \max_{a,b \in A} \|a - b\|_2 \qquad (2)$$

**Theorem 3.1.** *Let $Z_t$ and $Z_{t+1}$ denote the sets of visual token embeddings at layers $t$ and $t + 1$ of the vision encoder, respectively. The following statements hold*

*(i) The diameter of the token set is non-increasing across layers, i.e., $\mathbf{d}(Z_{t+1}) \leq \mathbf{d}(Z_t)$.*

*(ii) Assume that $Z_t$ can be partitioned into $M$ clusters $Z_t^m = \{z_t^j : j \in G^m\}$, where $\{G^m\}_{m=1}^M$ forms a partition of $\{1, \ldots, N\}$. For any $j \in G^m$ and $j' \in G^{m'}$ with $m \neq m'$, the attention weights satisfy $\varepsilon_l < a_t^{j',j} < \varepsilon_u$ with $\varepsilon_u \geq \varepsilon_l \geq 0$, indicating limited cross-cluster attention. Let $N_m = |G^m|$ and $A_m = N - N_m$. Then, the intra-cluster diameter at the next layer satisfies*

$$\mathbf{d}(Z_{t+1}^m) < (1 - A_m \varepsilon_l)^2 \mathbf{d}(Z_t^m) + A_m \varepsilon_u \mathbf{d}(Z_t)(A_m \varepsilon_u + 2)$$
$$(3)$$

Theorem 3.1 (proof in Appendix A.1) offers an asymptotic characterization of the clustering dynamics induced by uni-modal self-attention in the vision encoder. Specifically, Part (i) establishes that the overall diameter of the visual token embeddings does not increase across layers, reflecting an inherent contraction property of attention-based aggregation. Moreover, Part (ii) shows that when attention between distinct clusters diminishes, i.e., as the inter-cluster attention bounds $\varepsilon_u$ and $\varepsilon_l$ approach zero, the intra-cluster diameter is guaranteed to decrease, leading to progressively more compact visual token clusters over successive layers.

### 3.2. Token Grouping and Group-Level Representation

**Vision Token Clustering.** Motivated by the clustering behavior of self-attention representations analyzed in Section 3.1, we construct object-level visual token clusters from the *teacher* model. Given an input image, the teacher vision encoder produces a set of visual patch tokens $\{z_i^{\mathcal{T}}\}_{i=1}^{N_{\mathcal{T}}}$, where each token corresponds to an image patch and is associated with the normalized spatial coordinate of its patch center, $\mathbf{c}_i = (x_i, y_i) \in [0,1]^2$.

To jointly capture semantic similarity and spatial coherence, we define a pairwise distance matrix $\mathbf{D}^{\mathcal{T}} \in \mathbb{R}^{N_{\mathcal{T}} \times N_{\mathcal{T}}}$ as

$$D_{ij}^{\mathcal{T}} = 1 - \frac{\langle z_i^{\mathcal{T}}, z_j^{\mathcal{T}} \rangle}{\|z_i^{\mathcal{T}}\|_2 \|z_j^{\mathcal{T}}\|_2} + \lambda \|\mathbf{c}_i - \mathbf{c}_j\|_2,$$

where the first term measures cosine dissimilarity in representation space and the second term encodes spatial proximity in the image plane. By combining these two factors, the distance metric favors object-level grouping while avoiding the merging of semantically similar but spatially distant patches. We apply DBSCAN with this distance to $\mathbf{D}^{\mathcal{T}}$ to obtain teacher-side visual token clusters $\{C_m^{\mathcal{T}}\}_{m=1}^M$, while treating isolated or non-coherent tokens as noise.

Given the teacher clusters, we establish explicit correspondence to the *student* model based on spatial coverage. Each student visual token $z_j^{\mathcal{S}}$ is associated with a rectangular region $\mathcal{R}_j^{\mathcal{S}} \subset [0,1]^2$ in the normalized image plane, which represents the spatial extent of the image patch covered by the token, rather than a single point location. For each teacher cluster $C_m^{\mathcal{T}}$, the corresponding student cluster is defined as

$$C_m^{\mathcal{S}} = \{j \mid \exists i \in C_m^{\mathcal{T}} \text{ s.t. } \mathbf{c}_i \in \mathcal{R}_j^{\mathcal{S}}\}.$$

This region-based mapping allows multiple fine-grained teacher tokens to correspond to a single coarser student patch. Such a design is consistent with Theorem 3.1, which predicts increasing object-level compactness of visual token representations across layers. Further details of the teacher–student token mapping procedure are provided in the Appendix D.

**Text Span Construction.** For textual inputs, we construct linguistically grounded text spans rather than clustering tokens in representation space. We partition the selected Transformer layers into lower- and higher-level functional groups. At lower layers, contiguous tokens are grouped into complete word spans to preserve fine-grained lexical information, whereas at higher layers, spans correspond to phrase-level units such as noun phrases and verb phrases, capturing compositional semantics.

This hierarchical span construction yields progressively abstract textual representations and is shared by both the Teacher and Student. Figure 1 illustrates the resulting text spans and their interaction with visual clusters within the distillation objectives.

Span construction is implemented using a syntactic parser (e.g., spaCy) to extract spans from the input text. Each span represents a predefined set of tokens encoding a coherent semantic concept, enabling the Student to learn aggregated linguistic semantics rather than isolated word-level attributes.

**Unified Attention-Weighted Group Representation.** Given a pre-defined group of tokens $G_m = \{t_1, \ldots, t_{|G_m|}\}$, we construct a unified representation that summarizes the group semantics. Here, $G_m$ corresponds to a visual token cluster in the image modality or a linguistically constructed text span in the text modality, as defined in the preceding sections.

Instead of simple averaging, we employ an attention-weighted aggregation to allow more informative tokens within the group to contribute more strongly. Formally, at layer $l$, the representation of group $G_m$ is defined as

$$\mathrm{U}_{m,l} = \sum_{t \in G_m} w_{t,l}\, \mathbf{h}_{t,l}, \tag{4}$$

where $\mathbf{h}_{t,l}$ denotes the hidden state of token $t$ at layer $l$, and $w_{t,l}$ is its normalized importance weight.

The importance weights are derived from the Teacher model's self-attention. For each token $t \in G_m$, we accumulate the incoming attention it receives from other tokens within the same group across all attention heads:

$$\mathrm{Att}_{t,l} = \frac{1}{H} \sum_{h=1}^{H} \sum_{\substack{s \in G_m \\ s \neq t}} \alpha_{s \to t,l}^{(h)}, \tag{5}$$

where $\alpha_{s \to t,l}^{(h)}$ denotes the attention weight from token $s$ to token $t$ at head $h$ and layer $l$, computed with diagonal masking (and causal masking when applicable). The final weights are obtained by normalizing within the group:

$$w_{t,l} = \frac{\mathrm{Att}_{t,l}}{\sum_{r \in G_m} \mathrm{Att}_{r,l}}$$

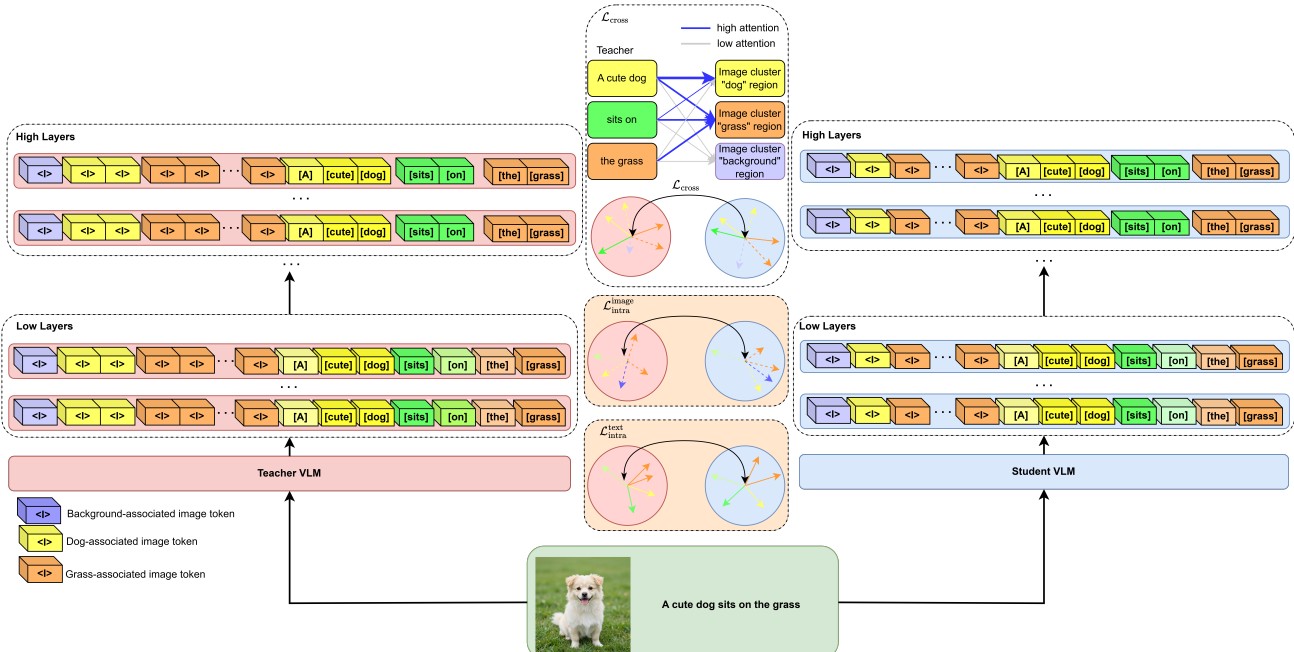

*Figure 1.* **Illustration of intra-modal and cross-modal distillation at the group level.** The Teacher guides the Student by preserving structural relationships within each modality (image clusters and text spans) as well as across modalities through cross-modal alignment.

## 3.3. Group-Level Distillation Objective

**Intra-modal Structural Alignment.** As illustrated in Figure 1, we enforce structural consistency between the Teacher and Student *within each modality* by preserving the relative geometry induced by the Teacher. For visual inputs, this geometry is defined over object-level image token clusters, whereas for textual inputs it is defined over linguistically constructed text spans.

Let $U_{i,l}^{T,\mu} \in \mathbb{R}^{d_T}$ and $U_{i,l}^{S,\mu} \in \mathbb{R}^{d_S}$ denote the representations of the $i$-th group at layer $l$ for the Teacher and Student, respectively, where $\mu \in \{\text{image}, \text{text}\}$ indexes the modality. Here, a group corresponds to an image cluster when $\mu = \text{image}$, and a text span when $\mu = \text{text}$.

For the visual modality, we align the intra-cluster geometry of the Student with that of the Teacher by matching pairwise distances between image clusters:

$$\mathcal{L}_{\text{intra}}^{\text{image}} = \sum_{l \in L_{\text{key}}} \frac{1}{|\mathcal{P}_l^{\text{image}}|} \sum_{(i,j) \in \mathcal{P}_l^{\text{image}}} \Delta_{i,j,l}^{\text{image}}, \qquad (6)$$

where $\mathcal{P}_l^{\text{image}}$ represents all pairs $(i,j)$ with $i \neq j$ of the clusters $i$ and $j$, and $d(\cdot, \cdot)$ denotes cosine distance, and we define

$$\Delta_{i,j,l}^{\text{image}} = \left( d(U_{i,l}^{S,\text{image}}, U_{j,l}^{S,\text{image}}) - d(U_{i,l}^{T,\text{image}}, U_{j,l}^{T,\text{image}}) \right)^2 \tag{7}$$

For the textual modality, we preserve the relative geometry among text spans induced by the Teacher in an analogous

manner:

$$\mathcal{L}_{\text{intra}}^{\text{text}} = \sum_{l \in L_{\text{key}}} \frac{1}{|\mathcal{P}_l^{\text{text}}|} \sum_{(m,n) \in \mathcal{P}_l^{\text{text}}} \Delta_{m,n,l}^{\text{text}}, \qquad (8)$$

where $\mathcal{P}_l^{\text{text}}$ represents all pairs $(m,n)$ with $m \neq n$ of the spans $m$ and $n$, and we define

$$\Delta_{m,n,l}^{\text{text}} = \left( d(U_{m,l}^{S,\text{text}}, U_{n,l}^{S,\text{text}}) - d(U_{m,l}^{T,\text{text}}, U_{n,l}^{T,\text{text}}) \right)^2 \tag{9}$$

The final intra-modal structural alignment loss is defined as

$$\mathcal{L}_{\text{intra}} = \mathcal{L}_{\text{intra}}^{\text{image}} + \mathcal{L}_{\text{intra}}^{\text{text}} \tag{10}$$

To investigate the theoretical insights of the image-intra and text-intra losses in preserving some characteristics from the teacher model, we develop the following theorem.

**Theorem 3.2.** *Assume that we have two collections of vectors $X = [x_i]_{i=1}^{n} \subset \mathbb{R}^{d_x}$ and $Y = [y_i]_{i=1}^{n} \subset \mathbb{R}^{d_y}$, with $d_x \leq d_y$. Suppose that the pairwise cosine distance are preserved, i.e., for all $i, j$, $1 - \frac{x_i^{\top} x_j}{\|x_i\|_2 \|x_j\|_2} = 1 - \frac{y_i^{\top} y_j}{\|y_i\|_2 \|y_j\|_2}$. Then, there exists a matrix $Q \in St(d_x, d_y)$ such that $\tilde{y}_i = Q\tilde{x}_i$ for all $i$, where $\tilde{x}_i = \frac{x_i}{\|x_i\|_2}$ and $\tilde{y}_i = \frac{y_i}{\|y_i\|_2}$ denote the $\ell_2$-normalized vectors. Here, $St(d_x, d_y) = \{R \in \mathbb{R}^{d_y \times d_x} \mid R^{\top} R = I_{d_x}\}$ denotes Stiefel manifold.*

We assume that the loss $\mathcal{L}_{\text{intra}}^{\text{image}}$ can be minimized to its true optimum of $0$. In this case, the two sets of vectors

$\left[U_{i,l}^{S,\text{image}}\right]_i$ and $\left[U_{i,l}^{T,\text{image}}\right]_i$ preserve pairwise cosine distances. By Theorem 3.2 (proof in Appendix A.2), there exists a matrix $Q^{\text{image}} \in \text{St}(d_S, d_T)$ with orthonormal columns (i.e., $(Q^{\text{image}})^\top Q^{\text{image}} = I_{d_S}$), which defines an isometric linear embedding, such that

$$\tilde{U}_{i,l}^{T,\text{image}} = Q^{\text{image}} \tilde{U}_{i,l}^{S,\text{image}}, \quad \forall i,$$

where $\tilde{U}_{i,l}^{S,\text{image}}$ and $\tilde{U}_{i,l}^{T,\text{image}}$ denote the normalized vectors of $U_{i,l}^{S,\text{image}}$ and $U_{i,l}^{T,\text{image}}$. Similarly, if $\mathcal{L}_{\text{intra}}^{\text{text}}$ is minimized to 0, there exists a matrix $Q^{\text{text}} \in \text{St}(d_S, d_T)$ such that

$$\tilde{U}_{m,l}^{T,\text{text}} = Q^{\text{text}} \tilde{U}_{m,l}^{S,\text{text}}, \quad \forall m,$$

with normalized version $\tilde{U}_{m,l}^{S,\text{text}}$ and $\tilde{U}_{m,l}^{T,\text{text}}$ of $U_{i,l}^{S,\text{text}}$ and $U_{i,l}^{T,\text{text}}$. Thus, minimizing both $\mathcal{L}_{\text{intra}}^{\text{image}}$ and $\mathcal{L}_{\text{intra}}^{\text{text}}$ enforces geometric alignment between teacher and student representations: the intrinsic relational structure of visual cluster and text spans is preserved across models up to an isometric transformation.

**Cross-modal Structural Alignment.** To preserve the relative geometry between visual and textual representations, we introduce a cross-modal distillation loss that aligns the image–text similarity structure of the Student with that of the Teacher. Unlike intra-modal alignment, this objective explicitly operates across modalities between image clusters and text spans, as illustrated in Figure 1.

At layer $l$, let $U_{i,l}^{T,\text{image}} \in \mathbb{R}^{d_T}$ and $U_{i,l}^{S,\text{image}} \in \mathbb{R}^{d_S}$ denote the representations of the $i$-th visual cluster in the Teacher and Student, respectively. Similarly, let $U_{j,l}^{T,\text{text}} \in \mathbb{R}^{d_T}$ and $U_{j,l}^{S,\text{text}} \in \mathbb{R}^{d_S}$ denote the representations of the $j$-th text span.

For each image–text pair $(i, j)$, we measure their cosine similarity in both models. The cross-modal structural alignment loss is defined as

$$\mathcal{L}_{\text{cross}} = \sum_{l \in L_{\text{key}}} \sum_{i \in \mathcal{C}_l^{\text{image}}} \sum_{j \in \mathcal{C}_l^{\text{text}}} \gamma_{i,j,l} \Big( d(U_{i,l}^{S,\text{image}}, U_{j,l}^{S,\text{text}})$$
$$- d(U_{i,l}^{T,\text{image}}, U_{j,l}^{T,\text{text}}) \Big)^2, \tag{11}$$

where $d(\cdot, \cdot)$ denotes cosine similarity.

The weight $\gamma_{i,j,l}$ reflects the semantic relevance of the image–text pair and is derived from the Teacher's cross-attention patterns. Specifically, we aggregate the cross-attention from all text tokens in text span $j$ to all visual tokens in visual cluster $i$ across attention heads, and normalize the result over all image–text pairs at layer $l$:

$$\gamma_{i,j,l} = \frac{\sum\limits_{t \in C_j^{\text{text}}} \sum\limits_{v \in C_i^{\text{img}}} \sum\limits_h \alpha_{t \to v,l}^{(h)}}{\sum\limits_{i' \in \mathcal{C}_l^{\text{img}}} \sum\limits_{j' \in \mathcal{C}_l^{\text{text}}} \sum\limits_{t \in C_{j'}^{\text{text}}} \sum\limits_{v \in C_{i'}^{\text{img}}} \sum\limits_h \alpha_{t \to v,l}^{(h)}} \tag{12}$$

This normalization emphasizes image-text pairs that are strongly associated in the Teacher's cross-modal geometry, while suppressing weak or noisy alignments.

We now investigate the role of cross-modal structural alignment in the following theorem.

**Theorem 3.3.** *Assume that the losses $\mathcal{L}_{\text{intra}}^{\text{image}}$ and $\mathcal{L}_{\text{intra}}^{\text{text}}$ can be optimized to 0. If the loss $\mathcal{L}_{\text{cross}}$ can be also optimized to 0 and span $\left\{ \tilde{U}_{i,l}^{S,\text{image}} \left( \tilde{U}_{m,l}^{S,\text{text}} \right)^T : i \in \mathcal{P}_l^{\text{image}}, m \in \mathcal{P}_l^{\text{text}} \right\} = \mathbb{R}^{d_S \times d_S}$, we then have $Q^{\text{text}} = Q^{\text{image}}$, where $Q^{\text{image}}$ and $Q^{\text{text}}$ are the linear isometries guaranteed by Theorem 3.2, with $Q^{\text{image}}, Q^{\text{text}} \in \text{St}(d_S, d_T)$, which preserve pairwise cosine similarities when mapping student representations to the teacher space.*

It is evident that Theorem 3.3 (proof in Appendix A.3) encourages the two modalities (i.e., images and texts) to be aligned more consistently by sharing the transformation matrices between the student and teacher models.

**Hidden Representation Alignment.** To further align fine-grained semantic content, we apply a feature-level distillation loss between corresponding *group-level* representations of the Teacher and Student, where a group denotes an image cluster for visual inputs or a text span for textual inputs.

Since the Student typically has a smaller hidden dimension ($d_S < d_T$), we introduce a learnable linear projector $W_l \in \mathbb{R}^{d_S \times d_T}$ to map Student representations into the Teacher feature space:

$$\hat{U}_{m,l}^S = U_{m,l}^S W_l. \tag{13}$$

The hidden alignment loss is defined as

$$\mathcal{L}_{\text{hid}} = \sum_{l \in L_{\text{key}}} \sum_m \left( 1 - \frac{(\hat{U}_{m,l}^S)^\top U_{m,l}^T}{\|\hat{U}_{m,l}^S\|_2 \|U_{m,l}^T\|_2} \right), \tag{14}$$

where $m$ indexes image clusters or text spans and $L_{\text{key}}$ denotes the selected layers used for distillation.

### 3.4. Combined Distillation Objective

We combine the standard multimodal embedding loss with the proposed hierarchical distillation objectives. Specifically, the student model is trained using the base loss together with our proposed structural alignment losses, and hidden-level alignment loss. The overall training objective is:

$$\mathcal{L}_{\text{total}} = \mathcal{L}_{\text{base}} + \lambda_{\text{struct}} \mathcal{L}_{\text{struct}} + \lambda_{\text{hid}} \mathcal{L}_{\text{hid}}, \tag{15}$$

where $\mathcal{L}_{\text{struct}} = \mathcal{L}_{\text{intra}} + \mathcal{L}_{\text{cross}}$, and $\lambda_{\text{struct}}$ and $\lambda_{\text{hid}}$ control the contributions of structural and hidden-level distillation, respectively.

*Table 1.* Multi-run quantitative comparison on CLS and VQA benchmarks using Precision@1 as the evaluation metric. For student methods, we report mean $\pm$ std over multiple runs and show only the aggregated rows. For each student backbone, the best and second-best mean Precision@1 scores are highlighted in **bold** and underline, respectively.

| Method | CLS | | | | | | VQA | | | | | | |
|---|---|---|---|---|---|---|---|---|---|---|---|---|---|
| | ImageNet-1K | N24News | HatefulMemes | VOC2007 | SUN397 | Avg | OK-VQA | A-OKVQA | DocVQA | InfoVQA | ChartQA | Visual7W | Avg |
| *B3-Qwen2-2B → FastVLM-0.5B* | | | | | | | | | | | | | |
| Teacher | 82.9 | 79.2 | 56.2 | 88.0 | 80.9 | 77.4 | 63.0 | 53.5 | 92.3 | 58.9 | 53.1 | 53.1 | 62.3 |
| SFT | 52.57 ± 0.29 | 68.90 ± 0.50 | 59.99 ± 0.06 | 78.23 ± 0.50 | 65.00 ± 0.26 | 64.93 ± 0.16 | 50.77 ± 0.10 | 50.30 ± 0.36 | 74.63 ± 0.55 | 34.53 ± 0.40 | 49.67 ± 0.25 | 46.37 ± 0.38 | 51.03 ± 0.28 |
| MSE | 52.90 ± 0.26 | 70.17 ± 0.40 | 58.33 ± 0.12 | 78.03 ± 0.15 | 65.17 ± 0.25 | 64.92 ± 0.07 | **52.63 ± 0.25** | 50.13 ± 0.31 | 74.20 ± 0.17 | 34.03 ± 0.25 | **50.93 ± 0.15** | 46.77 ± 0.35 | 51.45 ± 0.12 |
| RKD | 53.73 ± 0.32 | 70.40 ± 0.17 | 60.97 ± 0.21 | 79.00 ± 0.36 | 65.97 ± 0.21 | 66.17 ± 0.14 | 51.13 ± 0.25 | 51.13 ± 0.31 | 73.93 ± 0.29 | 35.10 ± 0.17 | 50.10 ± 0.36 | 46.53 ± 0.25 | 51.32 ± 0.07 |
| CKD | 53.13 ± 0.21 | 70.47 ± 0.23 | **61.00 ± 0.20** | 78.47 ± 0.32 | 66.97 ± 0.25 | 66.01 ± 0.08 | 51.87 ± 0.31 | 50.73 ± 0.38 | 74.83 ± 0.23 | 35.00 ± 0.40 | 49.97 ± 0.32 | 45.90 ± 0.36 | 51.38 ± 0.17 |
| EMO | 52.60 ± 0.20 | 68.27 ± 0.12 | 59.10 ± 0.20 | 80.03 ± 0.55 | 59.20 ± 0.17 | 63.84 ± 0.12 | 51.03 ± 0.15 | 49.60 ± 0.46 | 73.30 ± 0.20 | 34.03 ± 0.32 | 48.50 ± 0.50 | **47.47 ± 0.35** | 50.66 ± 0.13 |
| EM-KD | 53.20 ± 0.17 | 67.23 ± 0.12 | 59.20 ± 0.17 | 78.23 ± 0.21 | 63.33 ± 0.15 | 64.24 ± 0.05 | 51.17 ± 0.06 | 49.43 ± 0.23 | 77.17 ± 0.06 | 36.80 ± 0.10 | 47.47 ± 0.23 | 47.33 ± 0.15 | 51.56 ± 0.05 |
| **HieRD** | **55.93 ± 0.21** | **71.27 ± 0.38** | 60.83 ± 0.47 | **80.37 ± 0.25** | **67.13 ± 0.12** | **67.11 ± 0.06** | 52.60 ± 0.26 | **51.70 ± 0.10** | **77.57 ± 0.06** | **38.07 ± 0.21** | 50.07 ± 0.31 | 46.53 ± 0.31 | **52.76 ± 0.03** |
| *B3-Qwen2-2B → LLaVA-OneVision-0.5B* | | | | | | | | | | | | | |
| Teacher | 82.9 | 79.2 | 56.2 | 88.0 | 80.9 | 77.4 | 63.0 | 53.5 | 92.3 | 58.9 | 53.1 | 53.1 | 62.3 |
| SFT | 55.37 ± 0.25 | 66.90 ± 0.20 | 57.17 ± 0.15 | 83.97 ± 0.06 | 65.73 ± 0.25 | 65.83 ± 0.08 | 47.93 ± 0.31 | 43.37 ± 0.60 | 38.43 ± 0.15 | 21.33 ± 0.45 | 25.17 ± 0.40 | 43.80 ± 0.30 | 36.67 ± 0.28 |
| MSE | 55.07 ± 0.21 | 66.77 ± 0.55 | 56.10 ± 0.17 | 84.30 ± 0.26 | 66.97 ± 0.29 | 65.84 ± 0.18 | 48.07 ± 0.21 | 42.20 ± 0.30 | 37.90 ± 0.30 | 21.17 ± 0.21 | 24.73 ± 0.15 | 42.97 ± 0.40 | 36.17 ± 0.07 |
| RKD | 55.20 ± 0.17 | 67.03 ± 0.25 | 57.57 ± 0.45 | 83.60 ± 0.36 | 66.97 ± 0.15 | 66.07 ± 0.03 | 47.83 ± 0.25 | 43.60 ± 0.17 | 39.07 ± 0.42 | 21.57 ± 0.55 | 25.50 ± 0.26 | 43.80 ± 0.26 | 36.89 ± 0.12 |
| CKD | 56.67 ± 0.42 | 66.97 ± 0.31 | 58.47 ± 0.15 | 84.47 ± 0.47 | 68.17 ± 0.35 | 66.95 ± 0.24 | 51.13 ± 0.40 | 44.13 ± 0.21 | 47.17 ± 0.25 | 22.77 ± 0.21 | 31.27 ± 0.32 | 45.83 ± 0.25 | 40.38 ± 0.13 |
| EMO | 55.97 ± 0.15 | 66.20 ± 0.17 | 56.63 ± 0.21 | 83.33 ± 0.31 | 66.10 ± 0.26 | 65.65 ± 0.08 | 46.17 ± 0.17 | 43.53 ± 0.31 | 41.10 ± 0.17 | 20.83 ± 0.15 | 22.67 ± 0.40 | 42.97 ± 0.15 | 36.23 ± 0.13 |
| EM-KD | 53.43 ± 0.21 | 63.37 ± 0.06 | 53.33 ± 0.23 | 82.63 ± 0.15 | 65.17 ± 0.15 | 63.59 ± 0.01 | 44.67 ± 0.06 | 38.07 ± 0.21 | 36.77 ± 0.47 | 21.30 ± 0.72 | 22.43 ± 0.38 | 39.53 ± 0.23 | 33.79 ± 0.22 |
| **HieRD** | **63.27 ± 0.06** | **73.63 ± 0.15** | **59.27 ± 0.06** | **85.73 ± 0.06** | **68.50 ± 0.17** | **70.08 ± 0.05** | **52.43 ± 0.21** | **46.93 ± 0.15** | **49.23 ± 0.31** | **23.40 ± 0.20** | **34.60 ± 0.26** | **46.90 ± 0.20** | **42.25 ± 0.06** |

## 4. Experiments

### 4.1. Experimental Setup

**Datasets.** Following prior work, we evaluate our method on two multimodal tasks, image classification (CLS) and visual question answering (VQA) drawn from the MMEB (Massive Multimodal Embedding Benchmark) (Jiang et al., 2025). For classification, we report results on ImageNet-1K, N24News, HatefulMemes, VOC2007, and SUN397, spanning general-domain, news-related, and social-media scenarios. For VQA, we employ OK-VQA, A-OKVQA, DocVQA, InfographicsVQA (InfoVQA), ChartQA, and Visual7W, which require grounding textual queries in visual content. Dataset statistics are provided in Appendix C.3.

**Models.** We adopt B3 Qwen2-2B (Thirukovalluru et al., 2025) as the Teacher model, a well-finetuned variant of Qwen2VL-2B-Instruct (Wang et al., 2024), providing strong multimodal embedding quality and stable cross-modal alignment. For the Student, we consider two lightweight architectures: FastVLM-0.5B (Vasu et al., 2025) and LLaVA-OneVision-0.5B (Li et al., 2025).

**Implementation and Evaluation Details.** Both Teacher and Student models are fine-tuned using LoRA-based parameter-efficient training. All experiments follow the standard MMEB evaluation protocol (Jiang et al., 2025), and Precision@1 is used as the primary metric across all tasks. Detailed training configurations, LoRA settings, and hyperparameters are reported in Appendix C.2.

**Baselines.** We compare our method against a range of traditional and state-of-the-art knowledge distillation methods for vision–language models and large language models, including MSE, Relational KD (Park et al., 2019), Comparative KD (Wilf et al., 2025), EMO (Truong et al., 2025), and EM-KD (Feng et al., 2026). Further details on these baselines are provided in Appendix C.1.

### 4.2. Main Results

Table 1 reports multi-run results on CLS and VQA benchmarks, where all scores are evaluated using Precision@1. Overall, HieRD consistently improves compact student models across different student architectures, showing that hierarchical relational distillation provides an effective mechanism for transferring multimodal representation structure from a stronger teacher.

**Distillation to FastVLM-0.5B.** For the FastVLM-0.5B student, HieRD achieves the best Precision@1 on 7 out of 11 individual benchmarks, and obtains the highest average score on both CLS and VQA. On classification tasks, HieRD reaches an average Precision@1 of 67.11, improving over standard fine-tuning by 2.18 points and over the strongest distillation baseline by 0.94 points. It achieves the best result on 4 out of 5 classification datasets, including ImageNet-1K, N24News, VOC2007, and SUN397. These gains suggest that HieRD effectively transfers both fine-grained category-level semantics and broader scene-level visual knowledge.

On VQA benchmarks, HieRD obtains the best average Precision@1 of 52.76, outperforming the strongest baseline by 1.20 points. It achieves the best result on 3 out of 6 VQA datasets, namely A-OKVQA, DocVQA, and InfoVQA. The improvement is particularly clear on document-centric and image-specific reasoning tasks: HieRD improves DocVQA from 77.17 to 77.57 and InfoVQA from 36.80 to 38.07 compared with the strongest baseline. Although HieRD is not the best on every individual VQA dataset, its superior average Precision@1 indicates that hierarchical alignment provides a more balanced transfer of modality-specific reasoning signals.

**Distillation to LLaVA-OneVision-0.5B.** When distilling into LLaVA-OneVision-0.5B, the advantage of HieRD becomes even more pronounced. HieRD achieves the best

Precision@1 on all 11 individual benchmarks and also obtains the highest average score on both CLS and VQA. On classification tasks, it reaches an average Precision@1 of 70.08, improving over standard fine-tuning by 4.25 points and over the strongest distillation baseline by 3.13 points. HieRD consistently leads on all five classification datasets, with especially large gains on ImageNet-1K and N24News.

For VQA, HieRD achieves an average Precision@1 of 42.25, surpassing the strongest baseline by 1.87 points. It improves performance across all evaluated VQA datasets, including substantial gains on DocVQA, where Precision@1 increases from 38.43 under standard fine-tuning to 49.23. HieRD also improves InfoVQA from 21.33 to 23.40 and ChartQA from 25.17 to 34.60. These results show that HieRD is not only effective for classification-oriented representation transfer, but also beneficial for multimodal reasoning tasks that require stronger cross-modal alignment.

Overall, the Precision@1 results demonstrate that HieRD provides a robust and architecture-agnostic distillation strategy. The consistent gains across both FastVLM-0.5B and LLaVA-OneVision-0.5B further show that hierarchical representation alignment can reliably enhance compact vision-language embedding models across different student architectures.

## 5. Ablation Study

To analyze the contribution of each component in our framework, we conduct ablation studies using Precision@1 with the B3-Qwen2-2B teacher (Thirukovalluru et al., 2025) and the FastVLM-0.5B student (Vasu et al., 2025), focusing on (i) the effect of individual loss components, (ii) alternative designs for group-level representation aggregation, (iii) the sensitivity to DBSCAN clustering granularity, and (iv) training-time efficiency. Additional experimental results are provided in Appendix E.

Beyond performance-oriented ablations, we further conduct an empirical analysis to examine whether the cross-modal alignment properties predicted by Theorem 3.3 manifest in practice.

**Impact of Loss Components.** Table 2 evaluates the contribution of each loss component in HieRD using Precision@1. The full HieRD configuration achieves the best result on all three representative benchmarks, indicating that the proposed intra-modal, cross-modal, and hierarchical objectives provide complementary supervision.

Removing any single loss term consistently degrades performance. For instance, excluding $\mathcal{L}_{intra}$ reduces SUN397 from 67.2 to 66.2, while removing $\mathcal{L}_{cross}$ leads to clear drops on ImageNet-1K and VOC2007, from 56.0 to 54.7 and from 80.6 to 78.9, respectively. Similarly, removing $\mathcal{L}_{hid}$ also

lowers performance across all datasets, showing that hierarchical alignment is beneficial beyond direct intra- and cross-modal relational matching.

The degradation becomes more pronounced when two components are removed simultaneously. In particular, ImageNet-1K Precision@1 drops to 54.2 when both $\mathcal{L}_{intra}$ and $\mathcal{L}_{cross}$ are excluded, which is much closer to the baseline student performance of 52.9. These results suggest that each component captures a different aspect of teacher-student alignment, and combining them is important for robust representation transfer.

*Table 2.* Ablation results of HieRD under different loss component configurations, evaluated using Precision@1.

| Method | ImageNet-1K | VOC2007 | SUN397 |
|---|---|---|---|
| Student | 52.9 | 77.7 | 64.7 |
| HieRD (w/o $\mathcal{L}_{intra}$) | 55.2 | 80.5 | 66.2 |
| HieRD (w/o $\mathcal{L}_{cross}$) | 54.7 | 78.9 | 67.0 |
| HieRD (w/o $\mathcal{L}_{hid}$) | 55.7 | 80.4 | 66.8 |
| HieRD (w/o $\mathcal{L}_{intra}, \mathcal{L}_{cross}$) | 54.2 | 79.2 | 65.1 |
| HieRD (w/o $\mathcal{L}_{intra}, \mathcal{L}_{hid}$) | 53.4 | 78.7 | 65.4 |
| HieRD (w/o $\mathcal{L}_{cross}, \mathcal{L}_{hid}$) | 53.6 | 80.0 | 64.8 |
| **HieRD (full)** | **56.0** | **80.6** | **67.2** |

**Effectiveness of Attention-Weighted Aggregation.** We next validate the effectiveness of our representation aggregation design. As shown in Table 3, the proposed **Attention-Weighted** aggregation consistently outperforms the standard **Mean Pooling** baseline across all evaluated datasets. The gain is particularly clear on ImageNet-1K and SUN397, where Precision@1 improves from 53.7 to 56.0 and from 63.7 to 67.2, respectively.

These results indicate that simply averaging all tokens may dilute informative visual cues with less relevant or noisy tokens. In contrast, attention-weighted aggregation allows the model to assign larger weights to more discriminative tokens, producing more reliable group-level representations for hierarchical distillation. This supports our design choice of using attention-guided aggregation to construct semantic group representations.

*Table 3.* Comparison of aggregation strategies using Precision@1. We evaluate the impact of Mean Pooling versus Attention-Weighted aggregation.

| Aggregation Strategy | ImageNet-1K | VOC2007 | SUN397 |
|---|---|---|---|
| Mean | 53.7 | 80.1 | 63.7 |
| **Attention** | **56.0** | **80.6** | **67.2** |

**Sensitivity to DBSCAN Clustering Granularity.** We further analyze the sensitivity of HieRD to the DBSCAN `min_samples` hyperparameter, which controls the minimum number of neighboring samples required

to form a cluster. As shown in Table 4, the default setting `min_samples=8` achieves the best overall Precision@1 on CLS benchmarks. Using a smaller value, e.g., `min_samples=4`, leads to finer clusters and reduces the overall score from 67.16 to 66.12. In contrast, a larger value, e.g., `min_samples=12`, encourages coarser clusters and obtains an overall score of 66.58.

Conceptually, overly fine clustering makes the objective closer to token-wise distillation, while overly coarse clustering makes it closer to global alignment. The strongest performance is obtained when clusters capture mid-level visual entities, such as objects or object parts, which better matches the hierarchical structure targeted by HieRD.

*Table 4.* Ablation on the DBSCAN `min_samples` hyperparameter on CLS benchmarks, evaluated using Precision@1. The default setting in paper is `min_samples=8`.

| min_samples | ImageNet-1K | N24News | HatefulMemes | VOC2007 | SUN397 | Overall |
|---|---|---|---|---|---|---|
| 8 (default) | **56.0** | **71.7** | **60.3** | **80.6** | **67.2** | **67.16** |
| 4 | 55.3 | 70.7 | 58.6 | 79.6 | 66.4 | 66.12 |
| 12 | 55.8 | 71.2 | 59.0 | 80.3 | 66.6 | 66.58 |

**Training-Time Efficiency.** We compare the computational cost of HieRD with other distillation baselines in Table 5, reporting wall-clock time per iteration and average GPU memory usage under the configuration in Appendix C.2.

HieRD introduces a mild overhead over simpler objectives such as MSE and RKD due to hierarchical grouping and relational alignment. Nevertheless, its cost remains comparable to strong baselines: HieRD takes 3.6 seconds per iteration, close to EM-KD at 3.5 seconds, while using lower average VRAM than EM-KD (63.5GB vs. 64.8GB), even though EM-KD uses a smaller batch size. Together with the accuracy gains in Table 1, this indicates that HieRD offers a favorable accuracy–efficiency trade-off.

*Table 5.* Training-time efficiency comparison. We report wall-clock time per iteration and average GPU memory usage.

| Method | Time / Iter. (s) | Avg. VRAM (GB) |
|---|---|---|
| MSE | 2.9 | 57.7 |
| RKD | 3.0 | 60.1 |
| CKD | 3.3 | 60.5 |
| EMO | 3.3 | 61.2 |
| EM-KD | 3.5 | 64.8 |
| **HieRD** | 3.6 | 63.5 |

**Empirical Analysis of Theorem 3.3** In this subsection, we empirically validate the prediction of Theorem 3.3, which states that under sufficiently optimized intra-modal and cross-modal objectives, the learned linear isometries $Q^{\text{image}}$ and $Q^{\text{text}}$ should coincide. To examine this behavior, we track the orthogonality measure $(Q^{\text{image}})^\top Q^{\text{text}}$ during training.

As shown in Figure 2, the proposed orthogonality mea-

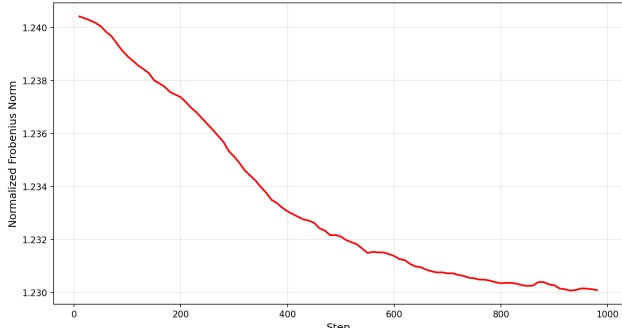

*Figure 2.* Visualization of the orthogonality measure between the learned isometries $Q^{\text{image}}$ and $Q^{\text{text}}$ across training. Lower values indicate closer alignment to the identity matrix.

sure exhibits a consistent decreasing trend during training. This behavior indicates that the two learned transformations $Q^{\text{image}}$ and $Q^{\text{text}}$ become increasingly aligned, suggesting convergence toward a shared isometric mapping. While not constituting a formal proof, this empirical observation provides supportive evidence for the hypothesis underlying Theorem 3.3, namely that sufficiently optimized intra-modal and cross-modal objectives encourage a unified cross-modal geometry.

Additional experimental details and analyses are provided in Appendix E.2.

## 6. Conclusion

We propose HieRD, a hierarchical distillation framework that overcomes the limitations of flat token-matching in vision–language models by aligning visual token clusters with phrase-level textual spans. This structure-aware approach effectively bridges architectural gaps between heterogeneous models, enabling robust knowledge transfer across different token counts and layer depths. Experimental results show that HieRD consistently outperforms existing baselines, producing efficient student models with high-quality semantic representations. By leveraging emergent cluster structures in transformers as stable alignment anchors, HieRD provides a scalable and comprehensive solution for distilling complex cross-modal correspondences.

## Acknowledgements

Trung Le was supported by the Air Force Office of Scientific Research under award number FA2386-25-1-4023 and the ARC Discovery Project grant DP250100262. Duc Anh Nguyen was supported by Hanoi University of Science and Technology (HUST) under Project No. T2024-PC-040.

## Impact Statement

This paper presents work whose goal is to advance research in vision–language modeling, particularly in the area of cross-modal distillation. Potential societal benefits include enabling more efficient multimodal models for deployment in resource-constrained environments. We do not identify any specific negative ethical consequences that must be highlighted here.

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

# A. Appendix

## A.1. Proof of Theorem 3.1

Throughout the proof, we follow the same notation and assumptions as in Section 3.1. For clarity, we first restate the theorem and the core definitions used in our analysis.

We define the diameter of a set of vectors $A \subset \mathbb{R}^D$ as the maximum Euclidean distance between any two elements in the set:

$$\mathbf{d}(A) = \max_{a,b \in A} \|a - b\|_2. \tag{16}$$

**Theorem A.1.** *Let $Z_t = \{\mathbf{z}_t^k\}_{k=1}^N$ and $Z_{t+1} = \{\mathbf{z}_{t+1}^k\}_{k=1}^N$ denote the sets of visual token embeddings at layers $t$ and $t+1$ of the vision encoder, respectively. Under the unimodal self-attention update with $W_V = I$. The following statements hold*

*(i) The diameter of the token set is non-increasing across layers, i.e., $\mathbf{d}(Z_{t+1}) \leq \mathbf{d}(Z_t)$.*

*(ii) Assume that $Z_t$ can be partitioned into $M$ clusters $Z_t^m = \{z_t^j : j \in G^m\}$, where $\{G^m\}_{m=1}^M$ forms a partition of $\{1, \ldots, N\}$. For any $j \in G^m$ and $j' \in G^{m'}$ with $m \neq m'$, the attention weights satisfy $\varepsilon_l < a_t^{j',j} < \varepsilon_u$ with $\varepsilon_u \geq \varepsilon_l \geq 0$, indicating limited cross-cluster attention. Let $N_m = |G^m|$ and $A_m = N - N_m$. Then, the intra-cluster diameter at the next layer satisfies*

$$\mathbf{d}(Z_{t+1}^m) < (1 - A_m \varepsilon_l)^2 \mathbf{d}(Z_t^m) + A_m \varepsilon_u \mathbf{d}(Z_t)(A_m \varepsilon_u + 2). \tag{17}$$

**Proof of Part (i).** With $h(\mathbf{a}, \mathbf{b}) = \|\mathbf{a} - \mathbf{b}\|_2$, we first note that

$$h(W_V a, W_V b) = \|W_V (a - b)\|_2 \leq \|W_V\|_2 \|a - b\|_2 \leq h(a, b).$$

Because the distance $h(\mathbf{a}, \mathbf{b}) = \|\mathbf{a} - \mathbf{b}\|_2$ is a convex function, we can apply Jensen's inequality. For any pair of tokens $j, j'$, we have:

$$\left\| \mathbf{z}_{t+1}^j - \mathbf{z}_{t+1}^{j'} \right\|_2 = \left\| \sum_{k=0}^N \mathbf{a}_t^{k,j} W_V \mathbf{z}_t^k - \sum_{k'=0}^N \mathbf{a}_t^{k',j'} W_V \mathbf{z}_t^{k'} \right\|_2$$

$$= h\left( W_V \sum_{k=0}^N \mathbf{a}_t^{k,j} \mathbf{z}_t^k, W_V \sum_{k'=0}^N \mathbf{a}_t^{k',j'} \mathbf{z}_t^{k'} \right)$$

$$\leq h\left( \sum_{k=0}^N \mathbf{a}_t^{k,j} \mathbf{z}_t^k, \sum_{k'=0}^N \mathbf{a}_t^{k',j'} \mathbf{z}_t^{k'} \right)$$

By applying Jensen's inequality for the convex function $h(\cdot, \cdot)$ twice, first for the first argument and then for the second, we obtain:

$$\left\| \mathbf{z}_{t+1}^j - \mathbf{z}_{t+1}^{j'} \right\|_2 \leq \sum_{k=0}^N \mathbf{a}_t^{k,j} h\left( \mathbf{z}_t^k, \sum_{k'=0}^N \mathbf{a}_t^{k',j'} \mathbf{z}_t^{k'} \right)$$

$$\leq \sum_{k=0}^N \sum_{k'=0}^N \mathbf{a}_t^{k,j} \mathbf{a}_t^{k',j'} h\left( \mathbf{z}_t^k, \mathbf{z}_t^{k'} \right).$$

Then, using the definition of the diameter $\mathbf{d}(Z_t) = \max_{k,k'} h(\mathbf{z}_t^k, \mathbf{z}_t^{k'})$ and the fact that $\sum_{k=0}^N \mathbf{a}_t^{k,j} = 1$, we have:

$$\left\| \mathbf{z}_{t+1}^j - \mathbf{z}_{t+1}^{j'} \right\|_2 \leq \sum_{k=0}^N \sum_{k'=0}^N \mathbf{a}_t^{k,j} \mathbf{a}_t^{k',j'} \mathbf{d}(Z_t)$$

$$= \left( \sum_{k=0}^N \mathbf{a}_t^{k,j} \right) \left( \sum_{k'=0}^N \mathbf{a}_t^{k',j'} \right) \mathbf{d}(Z_t)$$

$$= \mathbf{d}(Z_t).$$

Therefore, we reach $\mathbf{d}(Z_{t+1}) \leq \mathbf{d}(Z_t)$.

**Proof of Part (ii).** For any $j, j' \in G^m$, we derive as

$$\left\| \mathbf{z}_t^j - \mathbf{z}_t^{j'} \right\|_2 = \left\| \sum_{k=0}^N \mathbf{a}_t^{k,j} W_V \mathbf{z}_t^k - \sum_{k=0}^N \mathbf{a}_t^{k,j'} W_V \mathbf{z}_t^k \right\|_2 = h \left( \sum_{k=0}^N \mathbf{a}_t^{k,j} \mathbf{z}_t^k, \sum_{k=0}^N \mathbf{a}_t^{k,j'} \mathbf{z}_t^k \right)$$

$$\leq h \left( W_V \sum_{k=0}^N \mathbf{a}_t^{k,j} \mathbf{z}_t^k, W_V \sum_{k=0}^N \mathbf{a}_t^{k,j'} \mathbf{z}_t^k \right)$$

$$\leq \sum_{k=0}^N \sum_{k'=0}^N \mathbf{a}_t^{k,j} \mathbf{a}_t^{k',j'} h \left( \mathbf{z}_t^k, \mathbf{z}_t^{k'} \right)$$

$$\leq \sum_{k \in G^m} \sum_{k' \in G^m} \mathbf{a}_t^{k,j} \mathbf{a}_t^{k',j'} h \left( \mathbf{z}_t^k, \mathbf{z}_t^{k'} \right) + \sum_{k \in G^m} \sum_{k' \notin G^m} \mathbf{a}_t^{k,j} \mathbf{a}_t^{k',j'} h \left( \mathbf{z}_t^k, \mathbf{z}_t^{k'} \right)$$

$$+ \sum_{k \notin G^m} \sum_{k' \in G^m} \mathbf{a}_t^{k,j} \mathbf{a}_t^{k',j'} h \left( \mathbf{z}_t^k, \mathbf{z}_t^{k'} \right) + \sum_{k \notin G^m} \sum_{k' \notin G^m} \mathbf{a}_t^{k,j} \mathbf{a}_t^{k',j'} h \left( \mathbf{z}_t^k, \mathbf{z}_t^{k'} \right)$$

$$\leq \sum_{k \in G^m} \sum_{k' \in G^m} \mathbf{a}_t^{k,j} \mathbf{a}_t^{k',j'} \mathbf{d} \left( Z_t^m \right) + \sum_{k \in G^m} \sum_{k' \notin G^m} \mathbf{a}_t^{k,j} \mathbf{a}_t^{k',j'} \mathbf{d} \left( Z_t \right)$$

$$+ \sum_{k \notin G^m} \sum_{k' \in G^m} \mathbf{a}_t^{k,j} \mathbf{a}_t^{k',j'} \mathbf{d} \left( Z_t \right) + \sum_{k \notin G^m} \sum_{k' \notin G^m} \mathbf{a}_t^{k,j} \mathbf{a}_t^{k',j'} \mathbf{d} \left( Z_t \right).$$

We further manipulate as

$$\sum_{k \in G^m} \sum_{k' \in G^m} \mathbf{a}_t^{k,j} \mathbf{a}_t^{k',j'} \mathbf{d} \left( Z_t^m \right) = \mathbf{d} \left( Z_t^m \right) \sum_{k \in G^m} \mathbf{a}_t^{k,j} \sum_{k' \in G^m} \mathbf{a}_t^{k',j'}$$

$$= \mathbf{d} \left( Z_t^m \right) \left( 1 - \sum_{k \notin G^m} \mathbf{a}_t^{k,j} \right) \left( 1 - \sum_{k' \notin G^m} \mathbf{a}_t^{k',j'} \right)$$

$$< \mathbf{d} \left( Z_t^m \right) \left( 1 - A_m \epsilon_l \right)^2 \mathbf{d} \left( Z_t^m \right).$$

$$\sum_{k \in G^m} \sum_{k' \notin G^m} \mathbf{a}_t^{k,j} \mathbf{a}_t^{k',j'} \mathbf{d}(Z_t) = \mathbf{d}(Z_t) \sum_{k \in G^m} \mathbf{a}_t^{k,j} \sum_{k' \notin G^m} \mathbf{a}_t^{k',j'}$$

$$< A_m \epsilon_u \mathbf{d}(Z_t).$$

$$\sum_{k \notin G^m} \sum_{k' \in G^m} \mathbf{a}_t^{k,j} \mathbf{a}_t^{k',j'} \mathbf{d}(Z_t) = \mathbf{d}(Z_t) \sum_{k \notin G^m} \mathbf{a}_t^{k,j} \sum_{k' \in G^m} \mathbf{a}_t^{k',j'}$$

$$< A_m \epsilon_u \mathbf{d}(Z_t).$$

$$\sum_{k \notin G^m} \sum_{k' \notin G^m} \mathbf{a}_t^{k,j} \mathbf{a}_t^{k',j'} \mathbf{d}(Z_t) = \mathbf{d}(Z_t) \sum_{k \notin G^m} \mathbf{a}_t^{k,j} \sum_{k' \notin G^m} \mathbf{a}_t^{k',j'}$$

$$< A_m^2 \epsilon_u^2 \mathbf{d}(Z_t).$$

Finally, we arrive at

$$\left\| \mathbf{z}_t^j - \mathbf{z}_t^{j'} \right\|_2^2 < (1 - A_m \epsilon_l)^2 \mathbf{d}(Z_t^m) + A_m \epsilon_u \mathbf{d}(Z_t)(A_m \epsilon_u + 2).$$

$$\mathbf{d} \left( Z_{t+1}^m \right) < (1 - A_m \epsilon_l)^2 \mathbf{d}(Z_t^m) + A_m \epsilon_u \mathbf{d}(Z_t)(A_m \epsilon_u + 2).$$

## A.2. Proof of Theorem 3.2

Under the assumption, we have

$$1 - \frac{x_i^\top x_j}{\|x_i\|_2 \|x_j\|_2} = 1 - \frac{y_i^\top y_j}{\|y_i\|_2 \|y_j\|_2}$$

$$\tilde{x}_i^T \tilde{x}_j = \tilde{y}_i^T \tilde{y}_j, \forall i, j$$

Let denote $G_X = \left[\tilde{x}_i^T \tilde{x}_j\right]_{i,j=1}^n$ and $G_Y = \left[\tilde{y}_i^T \tilde{y}_j\right]_{i,j=1}^n$ as two Gram matrices, we then have $G_X = G_Y$. Moreover, we have $rank\,(G_X) = dim\,(span\,\{\tilde{x}_1, ..., \tilde{x}_n\}) = m$ and $rank\,(G_Y) = dim\,(span\,\{\tilde{y}_1, ..., \tilde{y}_n\}) = m$. Without loss of generality, we assume that $\{\tilde{x}_1, ..., \tilde{x}_m\}$ and $\{\tilde{y}_1, ..., \tilde{y}_m\}$ form basis of $span\,\{\tilde{x}_1, ..., \tilde{x}_n\}$ and $span\,\{\tilde{y}_1, ..., \tilde{y}_n\}$ respectively.

Now assume that $\tilde{x}_k = \sum_{i=1}^m \alpha_i \tilde{x}_i$ and $\tilde{y}_k = \sum_{i=1}^m \beta_i \tilde{y}_i$. It appears that

$$\begin{bmatrix} G_X[k,1] & ... & G_X[k,m] \end{bmatrix} = \tilde{x}_k^T \begin{bmatrix} \tilde{x}_1 & ... & \tilde{x}_m \end{bmatrix} = \alpha \begin{bmatrix} \tilde{x}_1^T \tilde{x}_1 & ... & \tilde{x}_1^T \tilde{x}_m \\ ... & ... & ... \\ \tilde{x}_m^T \tilde{x}_1 & ... & \tilde{x}_m^T \tilde{x}_m \end{bmatrix}$$

$$\begin{bmatrix} G_Y[k,1] & ... & G_Y[k,m] \end{bmatrix} = \tilde{y}_k^T \begin{bmatrix} \tilde{y}_1 & ... & \tilde{y}_m \end{bmatrix} = \beta \begin{bmatrix} \tilde{y}_1^T \tilde{y}_1 & ... & \tilde{y}_1^T \tilde{y}_m \\ ... & ... & ... \\ \tilde{y}_m^T \tilde{y}_1 & ... & \tilde{y}_m^T \tilde{y}_m \end{bmatrix}$$

Therefore, we have

$$\alpha \begin{bmatrix} \tilde{x}_1^T \tilde{x}_1 & ... & \tilde{x}_1^T \tilde{x}_m \\ ... & ... & ... \\ \tilde{x}_m^T \tilde{x}_1 & ... & \tilde{x}_m^T \tilde{x}_m \end{bmatrix} = \beta \begin{bmatrix} \tilde{y}_1^T \tilde{y}_1 & ... & \tilde{y}_1^T \tilde{y}_m \\ ... & ... & ... \\ \tilde{y}_m^T \tilde{y}_1 & ... & \tilde{y}_m^T \tilde{y}_m \end{bmatrix},$$

leading to $\alpha = \beta$ because two Gram matrices are identical and non-singular.

We extend the orthonormal set $\{e_j\}_{j=1}^m$ to a full orthonormal basis $\{e_1, \dots, e_{d_x}\}$ of $\mathbb{R}^{d_x}$. Similarly, since $d_y \geq d_x$, we extend $\{f_j\}_{j=1}^m$ to an orthonormal set $\{f_1, \dots, f_{d_x}\} \subset \mathbb{R}^{d_y}$, which is a subset of a full orthonormal basis of $\mathbb{R}^{d_y}$.

Define the matrices

$$E = [e_1, \dots, e_{d_x}] \in \mathbb{R}^{d_x \times d_x}, \qquad F = [f_1, \dots, f_{d_x}] \in \mathbb{R}^{d_y \times d_x}.$$

Since the vectors $\{e_k\}$ and $\{f_k\}$ are orthonormal, we have

$$E^\top E = I_{d_x}, \qquad F^\top F = I_{d_x}.$$

We define

$$Q = FE^\top \in \mathbb{R}^{d_y \times d_x}.$$

We first verify that $Q$ lies on the Stiefel manifold $St(d_x, d_y)$. Indeed,

$$Q^\top Q = (FE^\top)^\top (FE^\top) = EF^\top FE^\top = EI_{d_x}E^\top = EE^\top = I_{d_x}.$$

Thus, $Q \in St(d_x, d_y)$.

By construction of $Q = FE^\top$, we have

$$QE = FE^\top E = F,$$

which implies $Qe_k = f_k$ for all $k$. Therefore, the linear map $Q$ sends each basis vector $e_k$ to its corresponding vector $f_k$. We now prove that $Q\tilde{x}_k = \tilde{y}_k$.

Indeed, according to the Gram–Schmidt orthogonalization, we have $f_j^T \tilde{y}_i = e_j^T \tilde{x}_i$ for $i, j \leq m$. For $k \leq m$, we have $\tilde{x}_k = \sum_{i=1}^m e_i \left(\tilde{x}_k^T e_i\right)$. This follows

$$Q\tilde{x}_k = \sum_{i=1}^m Qe_i \left(\tilde{x}_k^T e_i\right) = \sum_{i=1}^m f_i \left(\tilde{y}_k^T f_i\right) = \tilde{y}_k$$

Finally, for $k > m$, we have

$$Q\tilde{x}_k = \begin{bmatrix} Q\tilde{x}_1 & ... & Q\tilde{x}_m \end{bmatrix} \alpha^T = \begin{bmatrix} \tilde{y}_1 & ... & \tilde{y}_m \end{bmatrix} \beta^T = \tilde{y}_k.$$

### A.3. Proof of Theorem 3.3

Under the assumptions, we have

$$d\left(U_{i,l}^{T,\text{image}}, U_{m,l}^{T,\text{text}}\right)^2 = d\left(U_{i,l}^{S,\text{image}}, U_{m,l}^{S,\text{text}}\right)^2$$

$$\left(\tilde{U}_{i,l}^{T,\text{image}}\right)^T \tilde{U}_{m,l}^{T,\text{text}} = \left(\tilde{U}_{i,l}^{S,\text{image}}\right)^T \tilde{U}_{m,l}^{S,\text{text}}.$$

This means that

$$\left(\tilde{U}_{i,l}^{S,\text{image}}\right)^T \left(Q^{\text{image}}\right)^T Q^{\text{text}} \tilde{U}_{m,l}^{S,\text{text}} = \left(\tilde{U}_{i,l}^{S,\text{image}}\right)^T \tilde{U}_{m,l}^{S,\text{text}}$$

$$\left(\tilde{U}_{i,l}^{S,\text{image}}\right)^T \left(\left(Q^{\text{image}}\right)^T Q^{\text{text}} - \mathbf{I}\right) \tilde{U}_{m,l}^{S,\text{text}} = 0.$$

Using the property of the trace, we gain

$$\text{trace}\left(\left(\left(Q^{\text{image}}\right)^T Q^{\text{text}} - \mathbf{I}\right)^T \tilde{U}_{i,l}^{S,\text{image}} \left(\tilde{U}_{m,l}^{S,\text{text}}\right)^T\right) = 0$$

$$\left\langle \left(Q^{\text{image}}\right)^T Q^{\text{text}} - \mathbf{I}, \tilde{U}_{i,l}^{S,\text{image}} \left(\tilde{U}_{m,l}^{S,\text{text}}\right)^T \right\rangle = 0.$$

This further implies that $\left(Q^{\text{image}}\right)^T Q^{\text{text}} - \mathbf{I}$ is orthogonal to span $\left\{\tilde{U}_{i,l}^{S,\text{image}} \left(\tilde{U}_{m,l}^{S,\text{text}}\right)^T : i, m\right\} = \mathbb{R}^{d \times d}$, implying that $\left(Q^{\text{image}}\right)^T Q^{\text{text}} - \mathbf{I} = \mathbf{0}$.

## B. Theoretical background of Relational KD

For a pair of samples, the distance-wise relational metric $\psi_D$ measures the Euclidean distance between their representations. The corresponding distillation loss $\mathcal{L}_{\text{RKD-D}}$ is defined over the set of all pairs of distinct examples $\mathcal{X}^2$ in a mini-batch:

$$\mathcal{L}_{\text{RKD-D}} = \sum_{(x_i, x_j) \in \mathcal{X}^2} l_\delta(\psi_D(t_i, t_j), \psi_D(s_i, s_j)),$$

where $t$ and $s$ denote the teacher and student representations, respectively. Furthermore, to capture higher-order geometric structures, an angle-wise relational potential $\psi_A$ is computed for each triplet of examples as the cosine of the angle formed at the $j$-th example:

$$\psi_A(t_i, t_j, t_k) = \cos \angle t_i t_j t_k = \langle \mathbf{e}^{ij}, \mathbf{e}^{kj} \rangle,$$

where the normalized direction vectors are given by $\mathbf{e}^{ij} = \frac{t_i - t_j}{\|t_i - t_j\|_2}$ and $\mathbf{e}^{kj} = \frac{t_k - t_j}{\|t_k - t_j\|_2}$. The angle-wise distillation loss is then formulated as:

$$\mathcal{L}_{\text{RKD-A}} = \sum_{(x_i, x_j, x_k) \in \mathcal{X}^3} l_\delta(\psi_A(t_i, t_j, t_k), \psi_A(s_i, s_j, s_k)).$$

In both objectives, $l_\delta$ denotes the Huber loss, which provides robustness to outliers and is defined as:

$$l_\delta(x, y) = \begin{cases} \frac{1}{2}(x - y)^2 & \text{for } |x - y| \leq 1, \\ |x - y| - \frac{1}{2} & \text{otherwise.} \end{cases}$$

The overall loss is defined as $L_{RKD} = L_{RKD-A} + L_{RKD-D}$.

# C. Experimental Details

## C.1. Baselines

We compare our method against several representative knowledge distillation baselines, which are summarized below.

- **MSE**: A point-wise distillation baseline where teacher and student embeddings are first projected into the same dimensional space, followed by alignment using Mean Squared Error.

- **Relational KD (RKD)** (Park et al., 2019): A structural distillation method that preserves inter-instance relationships by matching distance-wise and angle-wise potentials induced by the teacher.

- **Comparative KD (CKD)** (Wilf et al., 2025): A structural distillation approach designed for reduced-teacher-inference (RTI) settings. It replaces low-dimensional relations with high-dimensional comparisons between data points, enabling robust structural knowledge transfer under limited teacher supervision.

- **EMO** (Truong et al., 2025): A cross-tokenizer distillation framework originally proposed for text-only models. In our experiments, we follow the original training and distillation settings described in (Truong et al., 2025) and adapt them to the multimodal setting accordingly.

- **EM-KD** (Feng et al., 2026): A method for improving efficient multimodal large language models by addressing unbalanced vision tokens. Our implementation follows the pseudo-code and training procedure provided in the original paper.

## C.2. Implementation Details

**Models**    Our student models include **FastVLM-0.5B** (Vasu et al., 2025) and **LLaVA-OneVision-0.5B** (Li et al., 2025), two lightweight vision-language models. For the distillation process, we employ **B3** (Thirukovalluru et al., 2025) as the teacher model, which currently ranks among the state-of-the-art methods on the MMEB leaderboard. The teacher model (B3) is kept fixed during training. All models adopt the `eos` pooling strategy to produce fixed-size embeddings.

The detailed training configurations for each model are summarized in Table 6 and Table 7.

**Training and Evaluation**    The distillation process is conducted on two different benchmarks from the **MMEB** training benchmark, namely classification (CLS) and vision question answering (VQA). Parameter-efficient fine-tuning is applied via LoRA for all student models.

*Table 6.* Training configurations for all baseline methods, except EM-KD method.

| Settings | FastVLM-0.5B | LLaVA-OneVision-0.5B |
|---|---|---|
| Epoch | 1 | 1 |
| Learning Rate | $1 \times 10^{-4}$ | $1 \times 10^{-4}$ |
| Projector LR | $5 \times 10^{-4}$ | $5 \times 10^{-4}$ |
| Batch Size | 16 | 8 |
| LR Scheduler | Cosine | Cosine |
| Warmup Ratio | 0.03 | 0.03 |
| Weight Decay | 0.01 | 0.01 |
| LoRA Rank | 64 | 64 |
| LoRA Alpha | 64 | 64 |
| Image Resolution | 448 | 336 |

All experiments are implemented in **Python 3.11** using **PyTorch**. Training is performed on a **NVIDIA A100 SXM 80GB VRAM GPU**, and the `bf16` precision format is adopted to accelerate training and reduce memory consumption. To accommodate the high memory overhead of the EM-KD baseline within our available computational environment, we adapted its training configuration by reducing the effective batch size and the input image resolution.

*Table 7.* Training configurations for EM-KD method.

| Settings | FastVLM-0.5B | LLaVA-OneVision-0.5B |
|---|---|---|
| Epoch | 1 | 1 |
| Learning Rate | $1 \times 10^{-4}$ | $1 \times 10^{-4}$ |
| Projector LR | $5 \times 10^{-4}$ | $5 \times 10^{-4}$ |
| Batch Size | 8 | 4 |
| LR Scheduler | Cosine | Cosine |
| Warmup Ratio | 0.03 | 0.03 |
| Weight Decay | 0.01 | 0.01 |
| LoRA Rank | 64 | 64 |
| LoRA Alpha | 64 | 64 |
| Image Resolution | 448 | 128 |

**Hyperparameters.** The hyperparameters, $\alpha$, $\lambda_{\text{struct}}$ and $\lambda_{\text{hid}}$, are reported in Table 8. All reported values are selected based on preliminary validation experiments.

*Table 8.* Loss weighting coefficients used for different methods and model sizes.

| Datasets | FastVLM-0.5B | | | LLaVA-OneVision-0.5B | | |
|---|---|---|---|---|---|---|
| | $\alpha$ | $\lambda_{\text{struct}}$ | $\lambda_{\text{hid}}$ | $\alpha$ | $\lambda_{\text{struct}}$ | $\lambda_{\text{hid}}$ |
| CLS | 0.25 | 2.5 | 0.25 | 0.25 | 2.5 | 0.25 |
| VQA | 0.25 | 2.5 | 0.25 | 0.25 | 2.5 | 0.25 |

**Layer Mapping Configuration.** Table 9 summarizes the selected intermediate layers used for hierarchical distillation at the word and phrase levels. To accommodate differences in network depth between the Student and Teacher models, we align intermediate representations using a fixed proportional mapping rather than performing full-depth layer-wise matching. This design choice is motivated by the well-documented inter-layer redundancy in Transformer architectures, where adjacent layers often encode highly similar features.

Moreover, prior studies such as EMKD (Feng et al., 2026) and AlignKD (Feng et al., 2025) observe that, in multimodal Transformers, the most pronounced modality-specific transformations for visual representations injected into the LLM backbone occur at the earliest layers. These initial layers are responsible for projecting image features into the shared language-centric representation space, while deeper layers primarily refine already aligned semantic representations. Motivated by this observation, we explicitly include the first Transformer layer in our distillation set to ensure that early-stage cross-modal alignment is effectively transferred from the Teacher to the Student.

Concretely, let $N_S$ and $N_T$ denote the number of layers in the Student and Teacher models, respectively. Each selected Student layer index $l_S$ is mapped to its corresponding Teacher layer via proportional scaling, $l_T = \left\lfloor l_S \times \frac{N_T}{N_S} \right\rfloor$. Instead of exhaustively aligning all layers, we select a compact set of *key layers* $\mathcal{L}_{\text{key}}$ using a strided top-down strategy that prioritizes high-level semantic transitions closer to the output. Given a stride $k$ and a layer budget $M$, the selected Student indices are defined as $\mathcal{I}_S = \{N_S, N_S - k, \dots\}$ with $|\mathcal{I}_S| = M$. In practice, we use small strides (typically $k \in \{3, 4\}$), which effectively reduce redundancy while preserving a coherent representation trajectory for distillation.

The default layer selection in Table 9 is further examined in Table 12, where we ablate the number of selected phrase-level layers to evaluate the sensitivity of HieRD to this configuration.

### C.3. Dataset statistic

Table 10 presents the number of instances of each dataset from each task for the training and testing processes.

**Classification datasets**: The query consists of an instruction, an image, optionally accompanied by related text, while the target is the class label. The number of candidates equals the number of classes.

**Visual Question Answering datasets**: The query consists of an instruction, an image, and a piece of text as the question,

*Table 9.* Selected intermediate layers for hierarchical distillation under different tasks. CLS and VQA denote classification-oriented and question-answering-oriented supervision, respectively.

| Model | CLS | | VQA | |
|---|---|---|---|---|
| | **Word** | **Phrase** | **Word** | **Phrase** |
| FastVLM 0.5B | 0 | 18, 21, 24 | 0 | 18, 21, 24 |
| LLaVA-OneVision 0.5B | 0 | 18, 21, 24 | 0 | 18, 21, 24 |

*Table 10.* The statistics of the training datasets (Jiang et al., 2025)

| Meta-Task | Dataset | Query | Target | #Training | #Eval | #Candidates |
|---|---|---|---|---|---|---|
| Classification | ImageNet-1K | I | T | 100K | 1000 | 1000 |
| | N24News | I + T | I | 49K | 1000 | 24 |
| | HatefulMemes | I | T | 8K | 1000 | 2 |
| | VOC2007 | I | T | 8K | 1000 | 20 |
| | SUN397 | I | T | 20K | 1000 | 397 |
| VQA | OK-VQA | I + T | T | 9K | 1000 | 1000 |
| | A-OKVQA | I + T | T | 17K | 1000 | 1000 |
| | DocVQA | I + T | T | 40K | 1000 | 1000 |
| | InfographicVQA | I + T | T | 24K | 1000 | 1000 |
| | ChartQA | I + T | T | 28K | 1000 | 1000 |
| | Visual7W | I + T | T | 70K | 1000 | 1000 |

while the target is the answer. Each query has 1 ground truth and 999 distractors as candidates.

## D. Details of Vision Token Clustering and Teacher–Student Mapping

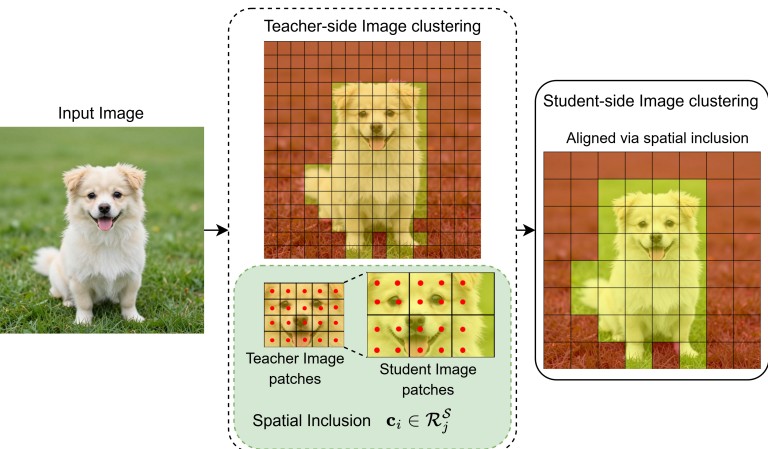

*Figure 3.* Illustration of the vision token clustering and teacher–student mapping procedure. Teacher-side visual tokens, represented by their patch centers $\mathbf{c}_i$, are first grouped into object-level clusters using DBSCAN. Each cluster is then associated with student visual tokens according to spatial inclusion within the corresponding patch regions $\mathcal{R}_j^{\mathcal{S}}$.

Figure 3 provides a schematic overview of the proposed vision token clustering and the subsequent teacher–student mapping procedure. In this section, we formally define the geometric quantities involved and describe how the mapping is computed in practice.

**Teacher token coordinates.** Let the teacher vision encoder process an input image of resolution $H_{\mathcal{T}} \times W_{\mathcal{T}}$, producing a set of visual patch tokens $\{z_i^{\mathcal{T}}\}_{i=1}^{N_{\mathcal{T}}}$. Each token corresponds to a patch centered at pixel location $(u_i^{\mathcal{T}}, v_i^{\mathcal{T}})$ in the teacher

input space. We define its normalized spatial coordinate as

$$\mathbf{c}_i = (x_i, y_i) = \left( \frac{u_i^{\mathcal{T}}}{W_{\mathcal{T}}}, \frac{v_i^{\mathcal{T}}}{H_{\mathcal{T}}} \right) \in [0,1]^2.$$

**Student patch regions.** Due to differences in vision preprocessors between the teacher and student models, the two encoders may operate on input images with different spatial resolutions. As a result, the student vision encoder processes images of resolution $H_{\mathcal{S}} \times W_{\mathcal{S}}$, which may differ from the teacher resolution $H_{\mathcal{T}} \times W_{\mathcal{T}}$. Each student visual token $z_j^{\mathcal{S}}$ corresponds to a rectangular region in the normalized image plane, defined as

$$\mathcal{R}_j^{\mathcal{S}} = [x_j^{\min}, x_j^{\max}) \times [y_j^{\min}, y_j^{\max}) \subset [0,1]^2,$$

where

$$x_j^{\min} = \frac{p_j^x}{W_{\mathcal{S}}}, \quad x_j^{\max} = \frac{p_j^x + w_j}{W_{\mathcal{S}}}, \quad y_j^{\min} = \frac{p_j^y}{H_{\mathcal{S}}}, \quad y_j^{\max} = \frac{p_j^y + h_j}{H_{\mathcal{S}}}.$$

Here $(p_j^x, p_j^y)$ denotes the top-left pixel coordinate of the student patch, and $(w_j, h_j)$ its width and height. By construction, the set of regions $\{\mathcal{R}_j^{\mathcal{S}}\}$ forms a partition of the normalized image plane, independent of the teacher resolution.

**Region-based cluster mapping.** Given a teacher-side visual token cluster $C_m^{\mathcal{T}}$, the corresponding student-side cluster is defined by spatial inclusion:

$$C_m^{\mathcal{S}} = \left\{ j \mid \exists i \in C_m^{\mathcal{T}} \text{ such that } \mathbf{c}_i \in \mathcal{R}_j^{\mathcal{S}} \right\}.$$

That is, a student token is associated with a teacher cluster if the center of at least one teacher patch belonging to that cluster falls within the spatial region represented by the student token. This formulation naturally supports many-to-one correspondence, where multiple fine-grained teacher tokens are mapped to a single coarser student patch, as illustrated in Figure 3.

# E. Additional Experimental Results

## E.1. Impact of Text Span Granularity

**HieRD** adopts a hierarchical span construction that assigns word-level spans to lower layers and phrase-level spans to higher layers, reflecting the progressive abstraction in Transformer representations. We compare this strategy with two static alternatives that apply a fixed granularity across all layers: word-level only and phrase-level only. As reported in Table 11, the hierarchical design consistently yields the strongest performance across all benchmarks. In particular, HieRD achieves accuracies of 56.0 on ImageNet-1K, 80.6 on VOC2007, and 67.2 on SUN397, exceeding the word-level baseline on all three datasets. While the phrase-level strategy performs competitively on VOC2007, it shows weaker results on ImageNet-1K and SUN397. These findings suggest that adapting text span granularity to layer depth provides a more effective supervision signal than using a single, fixed abstraction level.

*Table 11.* Ablation study on text span granularity using static and hierarchical strategies.

| Granularity Strategy | ImageNet-1K | VOC2007 | SUN397 |
|---|---|---|---|
| Word-level only | 54.9 | 79.7 | 66.5 |
| Phrase-level only | 55.2 | 80.6 | 66.4 |
| **HieRD (Ours)** | **56.0** | **80.6** | **67.2** |

## E.2. Additional Empirical Evidence for Theorem 3.3

This subsection provides additional empirical evidence supporting the assumptions and conclusions of Theorem 3.3. In particular, we examine whether the intra-modal and cross-modal objectives can be effectively optimized in practice, and whether their optimization leads to the geometric behaviors predicted by Theorems 3.2 and 3.3.

To enable rapid and controlled verification, we conduct a lightweight experiment using B3-Qwen2-2B (Thirukovalluru et al., 2025) as the teacher model and FastVLM-0.5B (Vasu et al., 2025) as the student model, evaluated on the VOC2007 dataset. All optimization settings and loss formulations are kept identical to those used in the main experiments.

We first examine the behavior of the intra-modal objectives. Figure 4 plots the evolution of the intra-image and intra-text losses during training. Both losses exhibit a clear decreasing trend and converge toward zero, indicating that the student representations increasingly preserve pairwise relationships within each modality. This empirical observation supports the assumption of Theorem 3.2, suggesting the existence of approximately orthogonal linear mappings $Q^{\text{image}}$ and $Q^{\text{text}}$ that align student representations with those of the teacher within each modality.

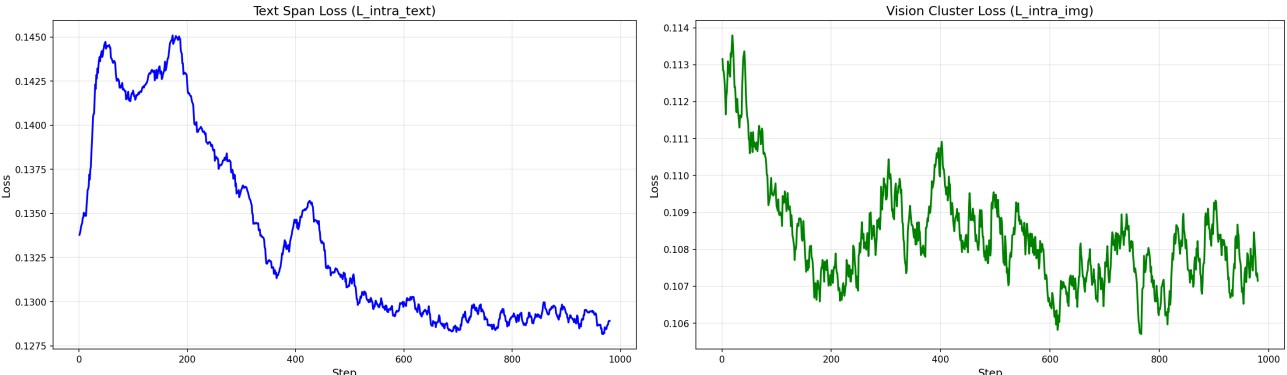

*Figure 4.* Evolution of intra-modal losses on VOC2007. Both intra-image and intra-text losses converge toward zero during training.

We further analyze the cross-modal objective by monitoring the cross-modal alignment loss. As shown in Figure 5, this loss also gradually decreases and converges over the course of training. Together with the convergence of the intra-modal losses, this behavior is consistent with the conditions of Theorem 3.3, suggesting that the learned transformations $Q^{\text{image}}$ and $Q^{\text{text}}$ become increasingly aligned during optimization. This observation aligns with the orthogonality analysis reported in the main paper, where $(Q^{\text{image}})^{\top} Q^{\text{text}}$ approaches the identity matrix during training.

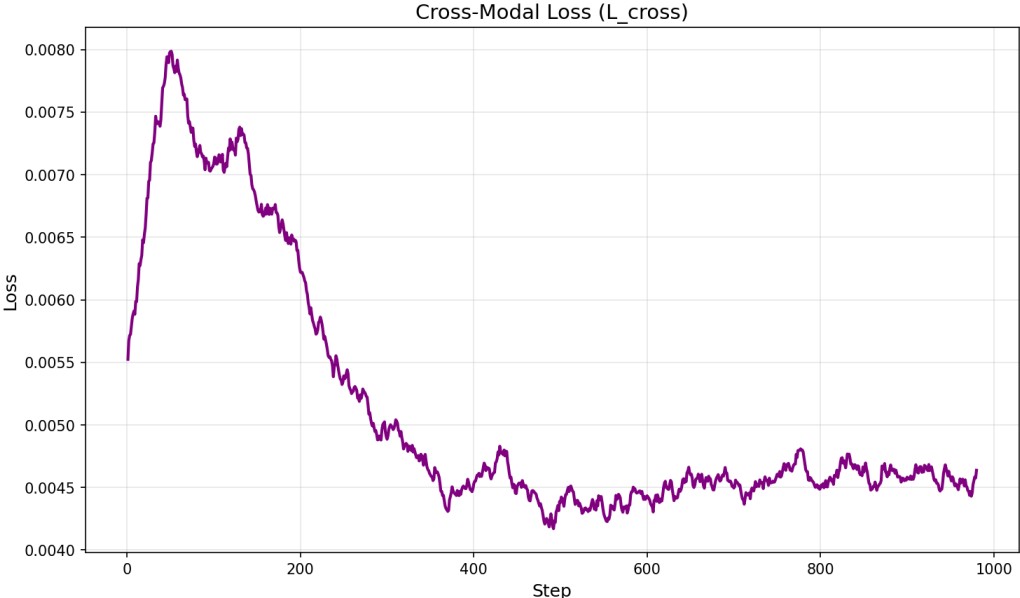

*Figure 5.* Evolution of the cross-modal loss on VOC2007. The loss decreases steadily, indicating improved alignment between image and text representations.

Although these results do not constitute a formal proof, they empirically validate the key assumptions underlying Theorem 3.3. In particular, they demonstrate that under sufficiently optimized intra-modal and cross-modal objectives, the learning dynamics naturally encourage convergence toward a unified cross-modal geometry, making the proposed theoretical framework practically meaningful.

## E.3. Ablation on Layer Selection

We further study the effect of selecting different intermediate layers for hierarchical distillation. In the main experiments, we use a compact four-layer configuration, consisting of one word-level layer and three phrase-level layers. To examine the sensitivity to this design choice, we compare it with a three-layer variant and a five-layer variant on CLS benchmarks using the B3-Qwen2-2B teacher and the FastVLM-0.5B student.

As shown in Table 12, the five-layer variant achieves the highest average Precision@1, improving the default four-layer configuration from 67.16 to 67.56. However, the gain is relatively modest, while adding one more phrase-level layer increases the amount of intermediate alignment during training. The three-layer variant performs slightly worse on average, suggesting that using too few intermediate layers may not sufficiently capture the hierarchical representation trajectory. Overall, the default four-layer setting provides a strong trade-off between performance and training cost, and is therefore used in the main experiments.

*Table 12.* Ablation on layer selection on CLS benchmarks, evaluated using Precision@1. The best and second-best results are highlighted in **bold** and underline, respectively.

| Config | ImageNet-1K | N24News | HatefulMemes | VOC2007 | SUN397 | Avg |
|---|---|---|---|---|---|---|
| 4 layers (default) | 56.0 | 71.7 | 60.3 | 80.6 | 67.2 | 67.16 |
| 3 layers | 54.2 | 71.5 | 60.9 | **81.9** | 65.7 | 66.84 |
| 5 layers | **56.6** | **71.8** | **61.0** | 80.9 | **67.5** | **67.56** |

Table 13 summarizes the corresponding layer selection configurations. In all variants, we keep the first Transformer layer for word-level alignment, since early layers are important for transferring cross-modal alignment from visual features into the language-centric representation space. We then vary the number of phrase-level layers selected from deeper layers. This design follows the same top-down selection principle used in the main method, while changing only the layer budget.

*Table 13.* Layer selection configurations used in the ablation study.

| Config | Word | Phrase |
|---|---|---|
| 4 layers (default) | 0 | 18, 21, 24 |
| 3 layers | 0 | 21, 24 |
| 5 layers | 0 | 15, 18, 21, 24 |

## E.4. Additional Results on Grounding Benchmarks

*Table 14.* Quantitative comparison on grounding benchmarks using Precision@1. The best student results are highlighted in **bold**.

| Method | IND | OOD | | | Avg |
|---|---|---|---|---|---|
| | MSCOCO | RefCOCO | RefCOCO-Mathching | Visual7W-Pointing | |
| Teacher (B3-Qwen2-2B) | 70.5 | 87.9 | 87.1 | 74.7 | 80.05 |
| SFT | 71.9 | 82.6 | 76.6 | 82.1 | 78.30 |
| CKD | 73.3 | 79.3 | 74.5 | 82.5 | 77.40 |
| RKD | 72.8 | 82.9 | **76.8** | 81.2 | 78.43 |
| EMO | 71.3 | 80.4 | 76.5 | 79.8 | 77.00 |
| EM-KD | 70.5 | 80.6 | 75.4 | 79.8 | 76.58 |
| **HieRD** | **73.4** | **83.0** | 76.4 | **82.7** | **78.88** |

The main experiments focus on CLS and VQA because they evaluate two complementary aspects of multimodal embedding models. CLS benchmarks measure global semantic discrimination and category-level representation structure, whereas VQA benchmarks emphasize fine-grained cross-modal interaction and reasoning. To further examine whether the improvements of HieRD extend beyond these two task groups, we additionally evaluate grounding benchmarks from MMEB, including both in-domain (IND) and out-of-domain (OOD) settings. We use B3-Qwen2-2B (Thirukovalluru et al., 2025) as the fixed teacher and FastVLM-0.5B (Vasu et al., 2025) as the student, following the same distillation setup as in the main experiments.

As shown in Table 14, HieRD achieves the best average Precision@1 among student methods, improving over the strongest baseline from 78.43 to 78.88. HieRD obtains the best result on MSCOCO, RefCOCO, and Visual7W-Pointing, and is competitive on RefCOCO-Matching. These results suggest that hierarchical relational distillation also benefits grounding-oriented evaluation, where the model needs to preserve spatially grounded visual-textual alignment rather than only global semantic matching.

## E.5. Additional Results on Other Retrieval Metrics

In the main paper, we follow the standard VLM2Vec evaluation protocol and use Precision@1 as the primary retrieval metric. To provide a more comprehensive evaluation, we additionally report Recall@3 (R@3) and Mean Reciprocal Rank (MRR).

As shown in Table 15, the conclusions remain consistent with the Precision@1 results. For the FastVLM-0.5B student, HieRD achieves the best performance on 4 out of 5 CLS datasets, all 6 VQA datasets, and the best overall average under both R@3 and MRR. For the LLaVA-OneVision-0.5B student, HieRD obtains the best or tied-best performance on all evaluated sub-datasets and also achieves the best overall average under both metrics. These results further demonstrate that the improvements of HieRD are stable across random seeds, student backbones, task types, and retrieval metrics.

*Table 15.* Multi-run evaluation on additional retrieval metrics. We report Recall@3 (R@3) and Mean Reciprocal Rank (MRR) as mean $\pm$ std over three runs.

**(a) CLS benchmarks (student: FastVLM-0.5B).**

| Method | ImageNet-1K | | N24News | | HatefulMemes | | VOC2007 | | SUN397 | | Avg | |
|---|---|---|---|---|---|---|---|---|---|---|---|---|
| | R@3 | MRR | R@3 | MRR | R@3 | MRR | R@3 | MRR | R@3 | MRR | R@3 | MRR |
| SFT | 74.07 ± 0.47 | 65.97 ± 0.25 | 88.73 ± 0.35 | 78.90 ± 0.26 | 100.00 ± 0.00 | 79.97 ± 0.25 | 93.13 ± 0.25 | **88.93 ± 0.38** | 85.73 ± 0.25 | 76.10 ± 0.26 | 88.33 ± 0.05 | 77.97 ± 0.13 |
| MSE | 74.10 ± 0.20 | 65.97 ± 0.15 | 88.97 ± 0.21 | 79.33 ± 0.15 | 100.00 ± 0.00 | 79.43 ± 0.15 | 92.80 ± 0.20 | 88.73 ± 0.15 | 85.93 ± 0.15 | 76.17 ± 0.15 | 88.36 ± 0.07 | 77.93 ± 0.03 |
| RKD | 74.27 ± 0.25 | 66.93 ± 0.23 | 88.23 ± 0.21 | 79.00 ± 0.36 | 100.00 ± 0.00 | 80.40 ± 0.46 | **93.87 ± 0.23** | 88.53 ± 0.21 | 85.77 ± 0.40 | 77.23 ± 0.35 | 88.43 ± 0.03 | 78.42 ± 0.11 |
| CKD | 74.13 ± 0.25 | 66.77 ± 0.35 | 89.27 ± 0.35 | 80.43 ± 0.25 | 100.00 ± 0.00 | 80.10 ± 0.44 | 93.83 ± 0.40 | 88.13 ± 0.21 | 86.27 ± 0.31 | 77.70 ± 0.46 | 88.70 ± 0.11 | 78.63 ± 0.08 |
| EMO | 74.63 ± 0.15 | 66.00 ± 0.10 | 87.33 ± 0.06 | 79.13 ± 0.06 | 100.00 ± 0.00 | 79.60 ± 0.10 | 90.37 ± 0.49 | 86.40 ± 0.44 | 85.70 ± 0.15 | 73.33 ± 0.12 | 87.56 ± 0.09 | 76.89 ± 0.12 |
| EM-KD | 72.97 ± 0.15 | 65.63 ± 0.06 | 88.00 ± 0.10 | 78.57 ± 0.06 | 100.00 ± 0.00 | 79.47 ± 0.12 | 89.67 ± 0.21 | 85.23 ± 0.06 | 86.17 ± 0.06 | 75.77 ± 0.15 | 87.36 ± 0.03 | 76.93 ± 0.01 |
| **HieRD** | **80.80 ± 0.36** | **70.47 ± 0.32** | **89.90 ± 0.00** | **81.73 ± 0.12** | **100.00 ± 0.00** | **80.63 ± 0.15** | 92.47 ± 0.12 | 87.37 ± 0.21 | **88.63 ± 0.29** | **79.67 ± 0.15** | **90.39 ± 0.01** | **79.97 ± 0.06** |

**(b) VQA benchmarks (student: FastVLM-0.5B).**

| Method | OK-VQA | | A-OKVQA | | DocVQA | | InfoVQA | | ChartQA | | Visual7W | | Avg | |
|---|---|---|---|---|---|---|---|---|---|---|---|---|
| | R@3 | MRR | R@3 | MRR | R@3 | MRR | R@3 | MRR | R@3 | MRR | R@3 | MRR | R@3 | MRR |
| SFT | 69.97 ± 0.51 | 62.33 ± 0.40 | 69.77 ± 0.45 | 61.67 ± 0.51 | 87.27 ± 0.61 | 82.03 ± 0.32 | 53.70 ± 0.46 | 45.87 ± 0.31 | 62.47 ± 0.21 | 58.53 ± 0.47 | 65.57 ± 0.35 | 56.93 ± 0.25 | 66.95 ± 0.23 | 62.67 ± 0.12 |
| MSE | 68.97 ± 0.38 | 60.10 ± 0.30 | 69.77 ± 0.40 | 60.60 ± 0.35 | 87.70 ± 0.36 | 82.20 ± 0.20 | 53.17 ± 0.31 | 45.33 ± 0.31 | 61.93 ± 0.42 | 58.70 ± 0.30 | 64.67 ± 0.61 | 57.27 ± 0.31 | 66.71 ± 0.11 | 61.77 ± 0.25 |
| RKD | 70.77 ± 0.32 | 63.23 ± 0.65 | 70.27 ± 0.50 | 61.03 ± 0.35 | 86.10 ± 0.26 | 83.93 ± 0.32 | 54.03 ± 0.35 | 45.77 ± 0.21 | 62.00 ± 0.26 | 59.03 ± 0.80 | 64.90 ± 0.30 | 56.73 ± 0.31 | 66.86 ± 0.22 | 62.81 ± 0.30 |
| CKD | 71.30 ± 0.36 | 63.87 ± 0.15 | 69.13 ± 0.51 | 60.47 ± 0.31 | 86.63 ± 0.42 | 84.13 ± 1.01 | 53.27 ± 0.50 | 45.53 ± 0.57 | 63.07 ± 0.31 | 57.87 ± 0.21 | 65.03 ± 0.25 | 57.27 ± 0.51 | 66.87 ± 0.26 | 62.81 ± 0.26 |
| EMO | 70.00 ± 0.30 | 63.00 ± 0.10 | 69.43 ± 0.49 | 60.20 ± 0.40 | 86.27 ± 0.15 | 82.67 ± 0.42 | 52.97 ± 0.15 | 46.67 ± 0.49 | 62.30 ± 0.26 | 57.63 ± 0.35 | 69.90 ± 0.46 | 57.63 ± 0.47 | 68.48 ± 0.11 | 61.30 ± 0.09 |
| EM-KD | 70.07 ± 0.06 | 62.70 ± 0.10 | 71.50 ± 0.20 | 62.93 ± 0.15 | 88.20 ± 0.10 | 83.57 ± 0.06 | 53.60 ± 0.36 | 48.33 ± 0.06 | 62.50 ± 0.17 | 57.07 ± 0.15 | 74.73 ± 0.06 | 63.00 ± 0.10 | 67.16 ± 0.13 | 64.89 ± 0.04 |
| **HieRD** | **72.60 ± 0.17** | **64.53 ± 0.15** | **73.50 ± 0.20** | **64.83 ± 0.06** | **89.40 ± 0.17** | **84.23 ± 0.06** | **55.53 ± 0.15** | **49.47 ± 0.15** | **64.13 ± 0.06** | **59.87 ± 0.06** | **76.60 ± 0.20** | **63.13 ± 0.15** | **69.17 ± 0.08** | **66.59 ± 0.03** |

**(c) CLS benchmarks (student: LLaVA-OneVision-0.5B).**

| Method | ImageNet-1K | | N24News | | HatefulMemes | | VOC2007 | | SUN397 | | Avg | |
|---|---|---|---|---|---|---|---|---|---|---|---|---|
| | R@3 | MRR | R@3 | MRR | R@3 | MRR | R@3 | MRR | R@3 | MRR | R@3 | MRR |
| SFT | 76.90 ± 0.10 | 69.40 ± 0.10 | 87.23 ± 0.15 | 78.10 ± 0.10 | 100.00 ± 0.00 | 79.23 ± 0.15 | 93.93 ± 0.15 | 88.53 ± 0.15 | 86.97 ± 0.15 | 77.00 ± 0.10 | 89.01 ± 0.03 | 78.45 ± 0.03 |
| MSE | 77.03 ± 0.15 | 69.57 ± 0.06 | 86.90 ± 0.10 | 77.80 ± 0.10 | 100.00 ± 0.00 | 79.30 ± 0.10 | 93.97 ± 0.15 | 88.20 ± 0.10 | 87.47 ± 0.06 | 77.13 ± 0.06 | 89.07 ± 0.01 | 78.40 ± 0.04 |
| RKD | 77.43 ± 0.21 | 68.90 ± 0.00 | 86.50 ± 0.17 | 78.07 ± 0.06 | 100.00 ± 0.00 | 79.50 ± 0.10 | 93.40 ± 0.17 | 87.93 ± 0.15 | 87.33 ± 0.12 | 77.57 ± 0.15 | 88.93 ± 0.04 | 78.39 ± 0.01 |
| CKD | 77.60 ± 0.26 | 69.33 ± 0.15 | 86.80 ± 0.20 | 77.83 ± 0.06 | 100.00 ± 0.00 | 78.70 ± 0.15 | 93.70 ± 0.10 | 87.80 ± 0.10 | 87.00 ± 0.10 | 78.07 ± 0.06 | 89.02 ± 0.08 | 78.35 ± 0.01 |
| EMO | 77.07 ± 0.15 | 69.07 ± 0.12 | 86.23 ± 0.15 | 77.40 ± 0.10 | 100.00 ± 0.00 | 79.00 ± 0.10 | 93.20 ± 0.20 | 87.77 ± 0.15 | 87.03 ± 0.21 | 77.10 ± 0.10 | 88.71 ± 0.09 | 78.07 ± 0.09 |
| EM-KD | 72.93 ± 0.12 | 65.50 ± 0.10 | 85.50 ± 0.10 | 75.80 ± 0.10 | 100.00 ± 0.00 | 76.57 ± 0.06 | 93.50 ± 0.26 | 88.77 ± 0.12 | 84.93 ± 0.15 | 76.40 ± 0.10 | 87.50 ± 0.11 | 76.61 ± 0.08 |
| **HieRD** | **84.97 ± 0.21** | **75.03 ± 0.06** | **91.27 ± 0.15** | **83.17 ± 0.06** | **100.00 ± 0.00** | **79.67 ± 0.06** | **94.47 ± 0.15** | **90.40 ± 0.10** | **88.93 ± 0.23** | **79.23 ± 0.15** | **91.93 ± 0.05** | **81.50 ± 0.07** |

**(d) VQA benchmarks (student: LLaVA-OneVision-0.5B).**

| Method | OK-VQA | | A-OKVQA | | DocVQA | | InfoVQA | | ChartQA | | Visual7W | | Avg | |
|---|---|---|---|---|---|---|---|---|---|---|---|---|
| | R@3 | MRR | R@3 | MRR | R@3 | MRR | R@3 | MRR | R@3 | MRR | R@3 | MRR | R@3 | MRR |
| SFT | 67.73 ± 0.12 | 60.17 ± 0.25 | 66.13 ± 0.38 | 56.57 ± 0.06 | 50.87 ± 0.15 | 47.17 ± 0.23 | 35.70 ± 0.35 | 31.60 ± 0.20 | 39.23 ± 0.32 | 35.43 ± 0.06 | 70.30 ± 0.12 | 59.87 ± 0.21 | 49.18 ± 0.16 | 50.28 ± 0.07 |
| MSE | 68.73 ± 0.32 | 60.17 ± 0.21 | 65.07 ± 0.06 | 56.13 ± 0.12 | 51.93 ± 0.21 | 47.10 ± 0.20 | 36.30 ± 0.17 | 31.83 ± 0.12 | 39.27 ± 0.38 | 35.53 ± 0.12 | 70.60 ± 0.20 | 59.70 ± 0.06 | 49.47 ± 0.12 | 50.23 ± 0.07 |
| RKD | 68.53 ± 0.15 | 60.33 ± 0.12 | 66.63 ± 0.15 | 57.03 ± 0.25 | 51.90 ± 0.10 | 47.93 ± 0.06 | 36.73 ± 0.21 | 32.43 ± 0.15 | 40.37 ± 0.32 | 35.80 ± 0.12 | 70.60 ± 0.10 | 59.77 ± 0.21 | 50.00 ± 0.15 | 50.69 ± 0.10 |
| CKD | 68.57 ± 0.15 | 60.37 ± 0.06 | 66.83 ± 0.25 | 57.10 ± 0.17 | 52.13 ± 0.32 | 47.80 ± 0.26 | 36.97 ± 0.21 | 32.43 ± 0.15 | 40.40 ± 0.26 | 35.93 ± 0.06 | 70.67 ± 0.06 | 59.70 ± 0.26 | 50.14 ± 0.12 | 50.72 ± 0.06 |
| EMO | 66.50 ± 0.10 | 58.37 ± 0.15 | 66.47 ± 0.15 | 57.00 ± 0.10 | 54.83 ± 0.25 | 49.37 ± 0.15 | 35.27 ± 0.15 | 31.30 ± 0.10 | 37.80 ± 0.20 | 33.70 ± 0.20 | 70.00 ± 0.20 | 59.37 ± 0.15 | 49.09 ± 0.03 | 49.96 ± 0.05 |
| EM-KD | 65.10 ± 0.17 | 56.67 ± 0.06 | 61.33 ± 0.12 | 52.87 ± 0.06 | 55.70 ± 0.00 | 48.97 ± 0.23 | 36.97 ± 0.06 | 33.50 ± 0.17 | 37.60 ± 0.35 | 33.47 ± 0.23 | 67.00 ± 0.17 | 55.67 ± 0.06 | 48.36 ± 0.04 | 48.74 ± 0.15 |
| **HieRD** | **70.13 ± 0.15** | **63.77 ± 0.12** | **69.23 ± 0.12** | **60.83 ± 0.15** | **65.33 ± 0.67** | **59.73 ± 0.06** | **40.70 ± 0.20** | **35.73 ± 0.21** | **49.73 ± 0.38** | **45.00 ± 0.20** | **73.67 ± 0.15** | **62.63 ± 0.06** | **56.69 ± 0.21** | **56.46 ± 0.09** |

## E.6. OOD Evaluation on CLS Benchmarks

We further evaluate the out-of-distribution (OOD) generalization ability of HieRD on additional CLS benchmarks. We use B3-Qwen2-2B (Thirukovalluru et al., 2025) as the fixed teacher and FastVLM-0.5B (Vasu et al., 2025) as the student, following the same distillation setting as in the main experiments. The evaluation includes Place365, ImageNet-A, ImageNet-R, ObjectNet, and Country211, which cover different types of distribution shifts such as natural scene variation, adversarial examples, rendition shifts, object-centric distribution changes, and geographical category shifts.

As shown in Table 16, HieRD achieves the best overall Precision@1 among student methods, improving over the strongest

baseline from 33.0 to 35.1. HieRD obtains the best result on ImageNet-A and Country211, and the second-best result on ImageNet-R and ObjectNet. Although EM-KD performs best on Place365 and ImageNet-R, HieRD provides the strongest average performance across all OOD datasets. These results suggest that hierarchical relational distillation improves not only in-domain performance, but also the robustness of compact vision-language embedding models under distribution shifts.

*Table 16.* OOD evaluation on CLS benchmarks using Precision@1. The best and second-best student results are highlighted in **bold** and underline, respectively.

| Method | Place365 | ImageNet-A | ImageNet-R | ObjectNet | Country211 | Avg |
|---|---|---|---|---|---|---|
| Teacher (B3-Qwen2-2B) | 44.5 | 50.7 | 90.4 | 72.0 | 26.2 | 56.8 |
| SFT | 29.9 | 31.7 | 51.8 | 44.7 | 8.0 | 33.2 |
| MSE | 28.9 | 30.8 | 48.9 | 43.2 | 7.8 | 31.9 |
| RKD | 29.5 | 23.3 | 52.6 | **49.4** | 7.2 | 32.4 |
| CKD | 30.4 | 29.8 | 48.6 | 47.1 | 8.3 | 32.8 |
| EMO | 29.7 | 26.1 | 54.7 | 44.2 | 7.5 | 32.4 |
| EM-KD | **30.5** | 22.3 | **58.4** | 46.5 | 7.1 | 33.0 |
| **HieRD** | 30.1 | **32.7** | 55.6 | 48.5 | **8.6** | **35.1** |

## E.7. Ablation with a Larger Teacher Model

We further evaluate HieRD under a stronger teacher setting by replacing the B3-Qwen2-2B teacher with B3-Qwen2-7B (Thirukovalluru et al., 2025), while keeping FastVLM-0.5B (Vasu et al., 2025) as the student. This setting introduces a larger teacher-student capacity gap and therefore provides a more challenging distillation scenario. In particular, a stronger teacher may produce richer but less directly transferable representation structures, making it harder for a compact student to imitate the teacher through standard feature- or relation-level objectives.

Table 17 summarizes the training settings used for this additional experiment. The same settings are used for HieRD and most baseline methods, while EM-KD uses a smaller batch size due to memory constraints. Unless otherwise specified, the training setup follows the same protocol as the main experiments.

*Table 17.* Training settings for the larger-teacher ablation. EM-KD uses a smaller batch size due to memory constraints.

| Setting | HieRD / Other Baselines | EM-KD |
|---|---|---|
| Teacher Model | B3-Qwen2-7B | B3-Qwen2-7B |
| Student Model | FastVLM-0.5B | FastVLM-0.5B |
| Epoch | 1 | 1 |
| Learning Rate | $1 \times 10^{-4}$ | $1 \times 10^{-4}$ |
| Projector LR | $5 \times 10^{-4}$ | $5 \times 10^{-4}$ |
| Batch Size | 16 | 8 |
| LR Scheduler | Cosine | Cosine |
| Warmup Ratio | 0.03 | 0.03 |
| Weight Decay | 0.01 | 0.01 |
| LoRA Rank | 64 | 64 |
| LoRA Alpha | 64 | 64 |
| Image Resolution | 448 | 448 |

As shown in Table 18, HieRD consistently outperforms standard fine-tuning and all prior distillation baselines on every evaluated CLS benchmark. It achieves an average Precision@1 of $67.02 \pm 0.05$, improving over SFT by 2.09 points and over the strongest distillation baseline, RKD, by 2.67 points.

Interestingly, using the larger B3-Qwen2-7B teacher does not necessarily improve the baseline distillation methods. Compared with the corresponding B3-Qwen2-2B teacher results in Table 1, most baselines become worse under the larger-teacher setting. For example, RKD decreases from 66.17 to 64.35, CKD decreases from 66.01 to 64.16, and EM-KD decreases from 64.24 to 61.28 on the average CLS score. Several distillation baselines also perform below the SFT student average of 64.93, suggesting that the larger teacher-student capacity gap can make knowledge transfer more difficult when the distillation objective does not explicitly account for hierarchical representation mismatch.

In contrast, HieRD remains robust under this stronger teacher setting. Its average Precision@1 stays nearly unchanged

*Table 18.* CLS results with a larger B3-Qwen2-7B teacher and FastVLM-0.5B student, evaluated using Precision@1. Student methods are reported as mean $\pm$ std over three runs. The best student results are highlighted in **bold**.

| Method | ImageNet-1K | N24News | HatefulMemes | VOC2007 | SUN397 | Avg |
|---|---|---|---|---|---|---|
| Teacher | 84.40 | 81.30 | 64.30 | 89.90 | 82.60 | 80.50 |
| SFT | $52.57 \pm 0.29$ | $68.90 \pm 0.50$ | $59.99 \pm 0.06$ | $78.23 \pm 0.50$ | $65.00 \pm 0.26$ | $64.93 \pm 0.16$ |
| MSE | $52.43 \pm 0.31$ | $67.03 \pm 0.12$ | $57.67 \pm 0.23$ | $79.90 \pm 0.17$ | $62.83 \pm 0.15$ | $63.97 \pm 0.10$ |
| RKD | $52.93 \pm 0.21$ | $67.23 \pm 0.25$ | $58.90 \pm 0.26$ | $79.40 \pm 0.20$ | $63.27 \pm 0.32$ | $64.35 \pm 0.08$ |
| CKD | $52.23 \pm 0.25$ | $67.80 \pm 0.30$ | $58.23 \pm 0.25$ | $79.00 \pm 0.17$ | $63.53 \pm 0.15$ | $64.16 \pm 0.22$ |
| EMO | $52.87 \pm 0.15$ | $67.27 \pm 0.40$ | $54.77 \pm 0.23$ | $80.10 \pm 0.56$ | $59.27 \pm 0.59$ | $62.85 \pm 0.14$ |
| EM-KD | $48.53 \pm 0.29$ | $65.00 \pm 0.17$ | $53.57 \pm 0.35$ | $77.67 \pm 0.32$ | $61.63 \pm 0.35$ | $61.28 \pm 0.11$ |
| **HieRD** | $\mathbf{55.23 \pm 0.15}$ | $\mathbf{71.70 \pm 0.20}$ | $\mathbf{61.53 \pm 0.31}$ | $\mathbf{80.60 \pm 0.17}$ | $\mathbf{66.03 \pm 0.31}$ | $\mathbf{67.02 \pm 0.05}$ |

compared with the B3-Qwen2-2B teacher setting in Table 1 (67.02 vs. 67.11), while still achieving the best result on all five CLS datasets. The improvements are especially clear on ImageNet-1K, N24News, and HatefulMemes, where HieRD improves over the strongest non-HieRD baseline by 2.30, 2.80, and 1.54 points, respectively. HieRD also improves VOC2007 from 80.10 to 80.60 and SUN397 from 65.00 to 66.03. These results indicate that the proposed hierarchical relational objectives are less sensitive to teacher-student capacity mismatch and can effectively preserve useful representation structure even when distilling from a substantially larger teacher.

