# OpenReview forum: "HieRD: Hierarchical Relational Distillation for Vision-Language Embedding Models"
_ICML.cc/2026/Conference — ICML 2026 regular_

### Official Review · Reviewer_LXVH · 2026-03-10

**Soundness:** 2
**Presentation:** 3
**Significance:** 3
**Originality:** 2
**Overall Recommendation:** 4
**Confidence:** 3

**Summary:**

This paper introduces HieRD (Hierarchical Relational Distillation), a framework designed to compress large Vision-Language Models (VLMs) into efficient embedding models for retrieval. Unlike traditional distillation that relies on flat token-wise matching, HieRD aligns visual token clusters (representing object-level entities) with hierarchical text spans (words at lower layers and phrases at higher layers). The framework employs intra-modal and cross-modal structural alignment losses to preserve the teacher's relational geometry across heterogeneous architectures. Evaluations on the MMEB benchmark show that HieRD consistently outperforms state-of-the-art distillation baselines across classification and VQA tasks.

**Compliance With Llm Reviewing Policy:**

Affirmed.

**Final Justification:**

After reading the response, my initial concerns have been fully resolved, and I am adjusting my score to a Weak Accept.

**Key Questions For Authors:**

1. The distance metric for DBSCAN relies on a spatial-semantic balance $\lambda$. How sensitive is the final retrieval performance to the choice of $\lambda$ and the DBSCAN $\epsilon$? If the clustering becomes too coarse or too fine, does the "hierarchical" benefit diminish?

2. Theorem 3.3 suggests that shared isometries $Q^{image}$ and $Q^{text}$ emerge under optimization. In cases where the teacher and student have vastly different latent space distributions (e.g., different training priors), does the convergence of the Frobenius norm shown in Figure 2 still hold as consistently?

**Limitations:**

yes

**Strengths And Weaknesses:**

Strengths:

1. The paper provides rigorous theoretical analysis to justify why transformer-based encoders naturally develop cluster structures and how isometric transformations can bridge the gap between teacher and student representation spaces.

Cons:

1. While the "spatial inclusion" mapping for visual clusters is intuitive, it assumes that the student's coarser patches can effectively inherit the semantics of the teacher’s fine-grained clusters. If a teacher cluster is semantically complex but spatially small, a single student patch might be forced to "overload" its representation, potentially leading to a semantic bottleneck that the current linear projector $W_l$ might not fully resolve.

2. HieRD uses DBSCAN to identify coherent visual clusters and treats isolated tokens as noise. However, in complex retrieval scenarios, "background" or non-coherent features often provide essential contextual cues. Discarding these tokens during group-level distillation might inadvertently weaken the student's ability to model global scene context, which is not explicitly addressed in the ablation studies.

---

> ### Author Rebuttal · Authors · 2026-03-30
>
> Thank you for the thoughtful comments and for recognizing the value of our theoretical analysis. We address each concern below.
>
> > # **[W1] Spatial inclusion may cause a semantic bottleneck when a small teacher cluster is mapped to a coarse student patch.**
>
> We agree this is an important concern. HieRD does not require a single student patch to directly copy a fine-grained teacher cluster. On the teacher side, clustering first aggregates multiple fine-grained tokens into a more stable **object-level representation** under the proposed spatial-semantic distance, which is also motivated by **Theorem 3.1**. Spatial inclusion is then used only to identify the student tokens most likely to correspond to that entity at a coarser resolution.
>
> Moreover, the linear projector does not simply compress information; it maps student features into the teacher latent space, which preserves sufficient capacity for alignment. HieRD is also applied across multiple layers rather than at a single depth, allowing **complex semantics to be absorbed progressively through the hierarchy** instead of being forced into one bottlenecked representation.
>
> > # **[W2] DBSCAN may ignore isolated/background tokens that still carry useful global context.**
>
> We would like to clarify that **HieRD does not discard background or isolated tokens from learning**. The DBSCAN-based grouping is introduced as a complementary hierarchical objective, not as a mechanism that removes non-clustered tokens from the model.
>
> First, isolated or background tokens remain fully involved in the forward pass. Even if they are not selected by the cluster-based structural objective, they still participate in token interaction throughout representation learning.
>
> Second, their contribution is indirect rather than explicitly supervised by a dedicated structural loss. Through self-attention, contextual information from background or non-coherent regions can still flow into the representations of clustered visual tokens and text-side features, which are optimized by other objectives in our framework. Therefore, such tokens are not directly matched at the structural level, but they are also not ignored: their information can still be propagated and absorbed into other supervised representations. We will clarify this distinction more explicitly in the revision.
>
> > # **[Q1] Sensitivity to the spatial-semantic balance and DBSCAN hyperparameters.**
>
> This is an important question. We do not rely on a single fixed $\varepsilon$ across settings; instead, **$\varepsilon$ is selected adaptively from the proposed distance matrix**, which makes clustering less sensitive than using a hand-picked constant threshold. To further clarify the effect of clustering granularity, we provide an additional sensitivity analysis in the table at **https://anonymous.4open.science/r/rebuttal-c9cc/exp_dbscan_sensitive.png**, including DBSCAN-related settings such as the minimum number of samples.
>
> Conceptually, if clustering is too fine, the objective approaches token-wise distillation; if it is too coarse, it approaches global alignment. The hierarchical benefit is strongest when clusters correspond to **mid-level visual entities such as objects or object parts**. We will clarify this guidance in the appendix together with the added sensitivity analysis.
>
> > # **[Q2] Does the convergence suggested by Theorem 3.3 still hold when teacher and student latent distributions differ substantially?**
>
> We agree that this is the critical regime to examine. Theorem 3.3 is primarily about **preserving relational geometry rather than exactly matching absolute feature values**. The shared isometry emerges because the student is encouraged to mimic the structure of the teacher latent space through the intra-modal and cross-modal objectives.
>
> Empirically, our setting already involves substantial mismatch: **FastVLM-0.5B has hidden dimension 896, whereas B3-Qwen2-2B has hidden dimension 1536, and the two models also differ in architecture**. Despite this, **Figures 2, 4 and 5** show that as optimization proceeds and the relevant losses decrease, the Frobenius norm and orthogonality-related quantities still converge stably. While this does not prove identical behavior for all possible prior mismatches, it suggests that **the optimization is robust enough to bridge meaningful distributional differences** when the relational objectives are sufficiently optimized. We will clarify this scope in the revision and note broader cross-teacher robustness as an important future direction.

---

> > ### Author Rebuttal · Reviewer_LXVH · 2026-04-03
> >
> > After reading the response, my initial concerns have been fully resolved, and I am adjusting my score to a Weak Accept.

---

> > > ### Author Response · Authors · 2026-04-03
> > >
> > > Thank you for the positive update to your evaluation. We are pleased that our clarifications addressed your concerns and appreciate the positive assessment of our work.

---

### Official Review · Reviewer_jgvf · 2026-03-12

**Soundness:** 3
**Presentation:** 3
**Significance:** 2
**Originality:** 3
**Overall Recommendation:** 3
**Confidence:** 3

**Summary:**

The paper addresses fine-grained knowledge preservation during VLM distillation for retrieval, proposing HieRD, a Hierarchical Representation Distillation framework. The motivation is well-grounded and the proposed remedy intuitive.

**Compliance With Llm Reviewing Policy:**

Affirmed.

**Key Questions For Authors:**

Q1 Figure 4: The Vision Cluster loss appears to converge less steadily than other loss components. Do the authors have a hypothesis for why this is?
Q2: As noted above, accuracy is an unusual choice for a retrieval paper. Could the authors report R@1, R@5, and R@10 in addition to (or instead of) accuracy? If there are specific reasons why accuracy was chosen

Comment: It would strengthen the paper to include a qualitative analysis to illustrate where the hierarchical distillation makes a difference.

**Limitations:**

The impact statement discusses only potential positive societal effects, risks are not discussed.

**Strengths And Weaknesses:**

Strengths
- The problem formulation is clear and well-motivated; The paper is well-written and easy to follow.
- The code is shared.
- The ablation study is extensive.


Weaknesses
- The impact statement discusses only potential positive societal effects. What are the risks? For instance, distilled VLMs deployed in retrieval systems could amplify representational biases present in the teacher model, or enable low-cost large-scale surveillance. The authors should discuss these possibilities.
- The results in the main tables appear to be based on single runs. Given that differences between methods are minor, it is unclear whether the reported gains are reliable. The authors should report means and standard deviations across at least three seeds, or conduct significance testing. This is especially important for the ablations in Table 2, where loss variants appear to perform comparably, it is hard to draw conclusions without knowing the variance.
- The authors evaluate model performance using classification accuracy. For retrieval, the standard metrics are Recall@k (R@1, R@5, R@10), Precision@K, and mAP@R. Accuracy is not a natural fit for ranking-based retrieval and may obscure meaningful differences in how well models surface relevant items. The authors should adopt retrieval-appropriate metrics, or at minimum justify the choice of accuracy in this context.

---

> ### Author Rebuttal · Authors · 2026-03-30
>
> Thank you for the careful reading and constructive suggestions. We address each point below.
>
> > # **[W1 and Limitation] Risks and broader societal impact.**
>
> We agree that the impact statement should discuss not only benefits but also risks. In the revision, we will explicitly add **three classes of concerns**. First, distilled VLMs may inherit or even amplify representational biases already present in the teacher, which can be harmful in retrieval and ranking settings. Second, **stronger and cheaper compact retrieval models could lower the barrier for large-scale surveillance or other privacy-invasive applications**. Third, KD may also transfer undesirable teacher behaviors, such as backdoor-triggered responses or other hidden failure modes. We view this as a broader challenge for KD methods, not only for HieRD. A practical future direction is to design **safer distillation procedures** that can better identify and suppress harmful teacher signals rather than simply compressing them. We will also mention the added training-time computation as a practical limitation.
>
> > # **[W2] Reliability of single-run results.**
>
> We agree that reporting only single runs is insufficient when the margins are modest. Due to limited preparation time before submission, the original tables reported single-run results. To address this concern, we have now repeated the experiments with **three random seeds and report the mean and standard deviation** in the table at **https://anonymous.4open.science/r/rebuttal-c9cc/exp_multirun.png**. The added results show that **the overall gains remain stable**, and they also make the conclusions in the **Experiments** section more reliable.
>
> > # **[W3 and Q2] Retrieval metrics beyond accuracy.**
>
> We agree that retrieval-oriented metrics are more informative in this setting. MMEB reformulates all four meta-tasks as ranking problems, where each query selects the correct target from a candidate pool. For retrieval, each query has one ground-truth item and 999 distractors, and the benchmark reports Precision@1 as the primary metric. In this setup, **Precision@1 is equivalent to Recall@1** because there is exactly one relevant target per query. Our implementation follows VLM2Vec [1], which reports Precision@1 across datasets.
>
> We also agree that additional metrics provide a fuller view of ranking quality. Therefore, we additionally report **R@3 and MRR** in the table at **https://anonymous.4open.science/r/rebuttal-c9cc/exp_additional_retrieval_metrics.png**. **R@3 measures whether the correct target appears in the top 3, while MRR reflects its ranking position**. The added results preserve the same overall conclusion.
>
> > # **[Q1] Why does the Vision Cluster loss converge less steadily in Figure 4?**
>
> Our hypothesis is that this behavior mainly comes from the data characteristics in that ablation, especially **VOC2007**. On the vision side, the images contain diverse object categories, layouts, and visual appearances, which makes **cluster-level alignment noisier and less uniform during training**. On the text side, the supervision pattern is much simpler and more repetitive, often following templates such as “Identify the object shown in the image.” As a result, **the text-related losses stabilize earlier, while the vision-cluster loss appears less smooth**. That said, although its convergence is slower, the overall trend is still downward, and it eventually approaches a value close to zero.
>
> Overall, we will strengthen the paper by expanding the risk discussion, reporting multi-seed results, clarifying the benchmark-specific use of P@1, and adding retrieval-oriented metrics.
>
> [1] VLM2Vec: Training Vision-Language Models for Massive Multimodal Embedding Tasks [ICLR 2025]

---

> > ### Author Rebuttal · Reviewer_jgvf · 2026-04-03
> >
> > I thank the authors for their thorough rebuttal and for providing additional experimental results.
> >
> > Regarding W2, while I appreciate the authors providing multi-seed results, inspection of the reported means and standard deviations reveals that the proposed method does not consistently outperform baselines across all three runs. Given that the majority of tables in the paper still report single-run results, it remains difficult to draw reliable conclusions about whether the observed gains are robust.
> >
> > Regarding W3/Q2, I appreciate the clarification but I remain confused: the paper refers to both "accuracy" and "Precision@k" at different points, and the main tables in the paper do not clearly label which metric is being reported. This ambiguity makes it difficult to situate the results within the broader literature. I would encourage the authors to use consistent, explicitly labelled metric names throughout the paper.
> >
> > I maintain my score.

---

> > > ### Author Response · Authors · 2026-04-03
> > >
> > > We address the remaining concerns on robustness and metric clarity below.
> > >
> > > **1. On robustness across random seeds (W2).**
> > >
> > > We respectfully disagree that the multi-seed results fail to support robust gains. In the rebuttal, we re-ran *all experiments from the main Table 1 using three different random seeds* and now report mean ± std for the aggregated rows only (Link: **https://anonymous.4open.science/r/rebuttal-c9cc/exp_multirun.png**).
> > >
> > > These results show that HieRD is consistently strong across both student backbones and both task groups:
> > >
> > > - Student **FastVLM-0.5B**, on task CLS: **HieRD** achieves the best average result, outperforming all the baseline models, with very small variance. It is also **best on 4/5 datasets: ImageNet-1K, N24News, VOC2007, and SUN397**.
> > > - Student **FastVLM-0.5B**, on task VQA: **HieRD** again achieves the best average result and still outperforms over all the other baselines with a significant gap. It is best on 3/6 datasets **(A-OKVQA, DocVQA, InfoVQA)** and **second-best on OK-VQA**, while still obtaining the strongest overall average.
> > > - Student **LLaVA-OneVision-0.5B**, on task CLS: HieRD is **best on all 5/5 datasets** and achieves the **best average (70.08 ± 0.05)**, outperforming the **strongest baseline CKD (66.95 ± 0.24)** by **3.13** points.
> > > - Student **LLaVA-OneVision-0.5B**, on task VQA: HieRD is again **best on all 6/6 datasets** and achieves the **best average (42.25 ± 0.06)**, surpassing the **strongest baseline CKD (40.38 ± 0.13)** by **1.87 points** (i.e., nearly 2 points).
> > >
> > > Therefore, the multi-run evidence is not a marginal or unstable effect: HieRD remains the strongest method in overall average for both CLS and VQA, under both student backbones, with consistently small standard deviations. We hope this directly addresses the concern that the conclusions might be driven by single-run noise.
> > >
> > > **2. On metric naming and clarity (W3/Q2).**
> > >
> > > We appreciate this suggestion and agree that the presentation can be made clearer. In the current paper, the primary metric in the main tables is Precision@1, following the standard VLM2Vec[1] evaluation protocol described in **Section 4.1 (Experimental Setup / Implementation and Evaluation Details)**. In our retrieval setting, each query is matched against 1 positive and 999 distractors, so Precision@1 is the protocol-consistent top-1 retrieval metric used throughout the main comparison.
> > >
> > > To further improve clarity, we will revise the final manuscript to make the metric labels explicit in every table header and use consistent terminology throughout. In particular, we will avoid any ambiguity between informal “accuracy” wording and the formal retrieval metric name Precision@1.
> > >
> > > Following the reviewer’s suggestion, we also added **Recall@3 (R@3)** and **Mean Reciprocal Rank (MRR)** in the rebuttal for a more granular analysis (Link: **https://anonymous.4open.science/r/rebuttal-c9cc/exp_additional_retrieval_metrics_all.png**). Importantly, the conclusions remain fully consistent under these additional retrieval metrics:
> > > - with **FastVLM-0.5B, HieRD** achieves the **best performance** on **4/5 CLS datasets, all 6/6 VQA datasets**, and the **best average** under both **R@3 and MRR**;
> > > - with **LLaVA-OneVision-0.5B, HieRD** achieves the **best performance** on **all sub-datasets** and **the overall average** under these additional metrics as well.
> > >
> > > Thus, both the original Precision@1 results and the newly added R@3/MRR evidence support the same conclusion: the gains of HieRD are robust and consistent across seeds, student backbones, task types, and retrieval metrics.
> > >
> > > We hope these additional clarifications and expanded multi-run / multi-metric results fully address your remaining concerns.
> > >
> > > [1] VLM2Vec: Training Vision-Language Models for Massive Multimodal Embedding Tasks [ICLR 2025]

---

### Official Review · Reviewer_2Qcv · 2026-03-13

**Soundness:** 3
**Presentation:** 2
**Significance:** 3
**Originality:** 3
**Overall Recommendation:** 4
**Confidence:** 3

**Summary:**

This paper proposes HieRD, a hierarchical distillation framework for vision-language embedding models. Instead of flat token-level matching, HieRD introduces (i) visual token clustering with semantic+spatial structure, (ii) phrase/span-level text grouping, and (iii) multi-granular distillation objectives (intra-modal, cross-modal, and hidden representation alignment). The method is evaluated on multimodal embedding and downstream VQA-style tasks, showing consistent gains over several KD baselines under compact student architectures.

**Compliance With Llm Reviewing Policy:**

Affirmed.

**Final Justification:**

The paper presents a reasonable and practically useful extension of multimodal distillation by introducing hierarchical relational alignment, with consistent empirical gains across settings. While the method is sound and well-motivated, its novelty is moderate and improvements are relatively incremental given the added training complexity. The rebuttal addressed most of my concerns by providing additional experiments and clarifying design choices, which strengthens the empirical support and clarity. However, these do not substantially change my overall assessment. I therefore maintain my original rating, with increased confidence.

**Key Questions For Authors:**

1. What is the training-time overhead compared to simpler KD baselines?

**Limitations:**

yes

**Strengths And Weaknesses:**

Strengths:
1. The paper is well motivated: distillation for multimodal embedding models is underexplored compared with reasoning-centric VLM work.
2. Moving from flat token alignment to hierarchical relational alignment is reasonable and practically meaningful.
3. Results are reported under two student backbones, with consistent improvements over strong KD baselines.

Weaknesses:
1. Novelty is moderate: The contribution is mainly a strong integration of known ingredients (clustering, relational distillation, span-level grouping), rather than a new distillation paradigm.
2. Theorems are derived under idealized assumptions (e.g., near-perfect loss minimization, isometric conditions). They are helpful intuitively but provide limited practical guarantees.
3. Performance may depend on clustering/parser/layer choices and hyperparameters; deeper sensitivity analysis would improve confidence.
4. Teacher diversity is limited: Most experiments use one teacher family; cross-teacher robustness is not fully established.

---

> ### Author Rebuttal · Authors · 2026-03-30
>
> Thank you for the feedback. We address each point below.
>
> > # **[W1]: Novelty is moderate.**
>
> We agree that HieRD is not a fully new distillation paradigm. Our goal is to address a specific gap in vision-language embedding distillation, where prior KD methods mainly target generative/reasoning behavior or use flat alignment objectives. In contrast, **HieRD is designed to preserve the hierarchical and relational semantic structure** needed by compact multimodal embeddings for transfer and generalization.
>
> Methods such as Align-KD [1] and EM-KD [2] improve alignment or affinity matching, but they do not explicitly group tokens into higher-level semantic units or align them across heterogeneous teacher-student architectures. HieRD contributes at this level through object-like visual clustering, phrase/span-level text grouping, and multi-granular relational alignment, preserving both intra-modal geometry and cross-modal correspondence. **Theorems 3.1--3.3** further motivate why these hierarchical anchors are meaningful and why the alignment direction is principled.
>
> > # **[W2]: Theorems rely on idealized assumptions.**
>
> We agree that the assumptions are idealized and do not constitute strict practical guarantees. Our intention is not to claim that all conditions hold exactly in practice, but to formalize the desired behavior of hierarchical distillation. In this sense, **the theory provides principled guidance for optimization**. Moreover, our ablation study and appendix analysis already provide empirical evidence that these conditions are at least approximately reflected during training. We will revise the paper to make this role clearer and better separate theoretical guidance from empirical validation.
>
> > # **[W3]: Sensitivity to clustering / parser / layer choices / hyperparameters.**
>
> We agree that further sensitivity analysis is valuable.
>
> First, the clustering settings mainly follow default configurations from the underlying implementation rather than heavy dataset-specific tuning. Likewise, text span grouping uses spaCy, a widely used parser, instead of a custom parsing module, which improves reproducibility.
>
> Second, we report an additional ablation with 3 / 4 / 5 layers on CLS, with results shown in the table at **https://anonymous.4open.science/r/rebuttal-c9cc/exp_layer_selection.png**. The results show that **3 layers are clearly weaker**, while 5 layers is only slightly better than 4, indicating limited gains beyond 4 layers. We therefore choose **4 layers as the best trade-off between performance and computational cost**.
>
> Third, although Eq. (15) contains multiple terms, in practice the method only requires tuning a small number of global coefficients, and we use one shared configuration across experiments rather than per-task retuning. We will clarify this in the revision.
>
> > # **[W4]: Teacher diversity is limited.**
>
> We agree that validation beyond one teacher family is important. Our current experiments already partially address heterogeneity because HieRD is evaluated on two substantially different student backbones, suggesting that it is not tied to one specific student design. More importantly, **HieRD is designed to reduce teacher-student mismatch by aligning relational structure rather than requiring direct token-wise correspondence**.
>
> To further strengthen this point, we report additional experiments using a larger teacher, B3-Qwen2-7B, with FastVLM-0.5B as the student. The added CLS results, shown in the table at **https://anonymous.4open.science/r/rebuttal-c9cc/exp_larger_teacher.png**, indicate that **HieRD remains stronger than RKD [3] and CKD [4] under the larger teacher-student gap**, while the baselines are more affected by the mismatch.
>
> > # **[Q1 / Limitation]: Training-time overhead.**
>
> This is an important practical question. HieRD does not modify the student architecture, so inference cost remains unchanged relative to the underlying student model. The extra cost appears only during training, due to the hierarchical grouping and relational alignment losses. We report the added training-time results in the table at **https://anonymous.4open.science/r/rebuttal-c9cc/exp_computation_comparision.png**, using the CLS task with FastVLM-0.5B as the student under the same training configuration as **Appendix C.2**. We will also state this trade-off explicitly in the limitations section: HieRD introduces moderate training overhead, but preserves deployment efficiency.
>
> Overall, we will revise the paper to clarify the novelty claim, better position the theory as principled guidance, and include the added analyses on layer sensitivity, teacher robustness, and training overhead.
>
> [1] Align-KD: Distilling Cross-Modal Alignment Knowledge for Mobile Vision-Language Model [CVPR 2025]
>
> [2] EM-KD: Distilling Efficient Multimodal Large Language Model with Unbalanced Vision Tokens [AAAI 2026]
>
> [3] Relational Knowledge Distillation [CVPR 2019]
>
> [4] Comparative Knowledge Distillation [WACV 2025]

---

> > ### Author Rebuttal · Reviewer_2Qcv · 2026-04-05
> >
> > I have carefully read the author rebuttal and appreciate the detailed responses and additional experimental results. The authors have addressed most of my concerns. The newly provided analyses strengthen the empirical support and clarify several previously unclear aspects. While some limitations remain, they are now better acknowledged and reasonably contextualized. Overall, the rebuttal improves my confidence in the work, and I will keep my original rating while updating my confidence to 4.

---

> > > ### Author Response · Authors · 2026-04-05
> > >
> > > Thank you for the time you invested in evaluating the rebuttal. We are glad that the additional analyses and clarified limitations improved your confidence in the work. Your decision to update your confidence to 4 is greatly appreciated.

---

### Official Review · Reviewer_bXmH · 2026-03-13

**Soundness:** 2
**Presentation:** 3
**Significance:** 2
**Originality:** 3
**Overall Recommendation:** 3
**Confidence:** 3

**Summary:**

This paper proposes HieRD, a Hierarchical Relational Distillation framework. HieRD groups visual tokens into semantically coherent clusters (representing object-level entities) and aligns them with linguistically grounded text spans (words at lower layers, phrases at higher layers). The framework employs multi-granular relational alignment objectives to preserve intra-modality and cross-modal structures. Experimental results on the Massive Multimodal Embedding Benchmark (MMEB) demonstrate that HieRD outperforms strong distillation baselines like RKD and EM-KD across two different student architectures.

**Compliance With Llm Reviewing Policy:**

Affirmed.

**Final Justification:**

**Soundness.** The paper proposes a hierarchical distillation framework for multi-modal embedding models with supporting theory and experiments. However, why the ingredients in the objective are meaningful is only explained by the theorems but not supported by experimental observations. Moreover, the choice of teacher model family is limited.

**Originality.** The hierarchical alignment of visual clusters with multi-level text units is novel in this specific setting.

**Significance.** While efficient multimodal embedding models are important, the limited performance improvements and increased complexity reduce practical impact. Generalization to other teacher models remains unclear.

**Clarity.** The paper is clear and well-structured.

Regarding my concerns:

W1: While the author argued that a shared coefficient setting is used for both tasks and students, it is unknown whether this choice generalizes to other settings and how heavy the tuning burden would be.

W2: Generalization beyond B3-Qwen2-2B is partially addressed by additional results of a larger teacher model, but not from a new family.

W3: For the additional results, improvements are relatively small and the evaluation variance is not provided.

Overall: A novel idea with implementation, but limited empirical gains and insights.

Final recommendation: I maintain my original score.

**Key Questions For Authors:**

1. What is the motivation of choosing CLS and VQA for evaluation and how does HieRD perform on the other two subsets of MMEB, retrieval and grounding?
2. How does the students' computation cost and performance compare to other non-KD multimodal embeddings models?
3. Was there a specific ablation on the number of key layers, and does more layers always lead to better performance, or is there a point of diminishing returns?

**Limitations:**

A discussion of the limitation of the method (e.g., computational overhead and applicability) can be added to improve the paper.

**Strengths And Weaknesses:**

Strengths:
1. The paper develops light-weight (0.5B) multimodal embedding models through knowledge distillation (KD), which outperforms SFT on small models.
2. The proposed KD loss shows gains in VQA and classification tasks compared with previous KD losses, particularly in document-centric tasks like DocVQA.
3. The method design is supported by the theoretical analysis.

Weaknesses:
1. The proposed training objective involves five loss components (in formula 15) and thus requires careful hyperparameter tuning of the loss weights, which increases the complexity of the method.
2. The teacher model choice is restricted to B3-Qwen2-2B. It is unclear whether the hyperparameter choices and the conclusions generalize to teacher models with a different backbone or a larger size.
3. The generalizability of the method is not reported in the paper, since it does not evaluate the models on OOD subsets of the CLS and VQA categories provided by the MMEB benchmark.

---

> ### Author Rebuttal · Authors · 2026-03-30
>
> Thank you for the constructive feedback. We address each concern below.
>
> > # **[W1] Complexity of the objective / hyperparameter tuning.**
>
> We agree this is an important practical concern. Although Eq. (15) contains five loss terms, HieRD only introduces **three tuned coefficients in practice:** $\alpha$, $\lambda_{\text{struct}}$, and $\lambda_{\text{hid}}$. More importantly, we do not tune them separately for each task or student. Instead, we find **one shared setting** that works across both CLS and VQA, and across both student backbones (FastVLM-0.5B and LLaVA-OneVision-0.5B), as reported in Table 6. In this sense, **the practical tuning burden is limited** and does not require task-specific rebalancing.
>
> > # **[W2] Generalization beyond B3-Qwen2-2B.**
>
> We agree that validation beyond one teacher is important. Our current experiments already partially address architecture mismatch, since HieRD is evaluated on two substantially different student backbones, suggesting that it is not tied to one specific student design. More importantly, HieRD is explicitly designed to reduce teacher-student mismatch by aligning relational structure rather than requiring direct token-wise correspondence. To further strengthen this point, we add experiments with a larger teacher, B3-Qwen2-7B, and FastVLM-0.5B as the student. The added CLS results in the table at **https://anonymous.4open.science/r/rebuttal-c9cc/exp_larger_teacher.png** show that **HieRD remains stronger than RKD [1] and CKD [2] under the larger teacher-student gap**, while the baselines are more affected by the increased mismatch.
>
> > # **[W3] OOD generalization.**
>
> This is a fair concern. We further report OOD evaluations on MMEB for the CLS task, compared with baseline KD methods, in the table at **https://anonymous.4open.science/r/rebuttal-c9cc/exp_OOD_Evaluation.png**. Although HieRD does not outperform the baselines on every individual OOD dataset, it remains competitive on the few cases where it is slightly lower and **achieves a better average score across the OOD benchmarks overall**. We believe this suggests stronger aggregate robustness under distribution shift, while also acknowledging that OOD transfer remains challenging for both multimodal training and KD. We will clarify this limitation and position improved OOD robustness as an important future direction.
>
> > # **[Q1] Why CLS and VQA? What about retrieval and grounding?**
>
> We selected CLS and VQA because they stress two complementary aspects central to our claim. CLS evaluates global semantic discrimination and whether distilled embeddings preserve category-level structure; VQA evaluates fine-grained cross-modal interaction and reasoning. Together, they provide a strong test of whether hierarchical relational distillation improves both coarse- and fine-grained multimodal alignment.
>
> For the remaining MMEB subsets, we additionally conduct experiments on grounding and evaluate its OOD datasets; some results are shown in the table at **https://anonymous.4open.science/r/rebuttal-c9cc/exp_grounding.png**. For retrieval, we agree it is highly relevant. However, because retrieval evaluation is relatively large-scale, we have not yet been able to complete a reliable full evaluation under our current computational constraints. We will clarify this scope limitation in the paper rather than overclaiming.
>
> > # **[Q2 + Limitation] Computation cost vs. KD multimodal embedding models.**
>
> Our goal is to improve the quality of small student models through KD, rather than pursue SOTA by scaling model size or introducing heavier inference-time designs. **HieRD does not modify the student architecture, so inference cost remains unchanged** relative to the underlying student backbone. The extra cost appears only during training, due to the hierarchical grouping and relational alignment losses. To make this comparison concrete and fair, we include the added computational cost and efficiency comparison in the table at **https://anonymous.4open.science/r/rebuttal-c9cc/exp_computation_comparision.png**, using the CLS task with FastVLM-0.5B as the student under the same training configuration as **Appendix C.2**. We believe this is a favorable trade-off for deployment-oriented small models: **training becomes moderately heavier, while inference remains unchanged.**
>
> > # **[Q3] Ablation on the number of key layers.**
>
> We agree this is important and add an ablation with 3 / 4 / 5 key layers on CLS, with results shown in the table at **https://anonymous.4open.science/r/rebuttal-c9cc/exp_layer_selection.png**. The results show that **3 layers are clearly weaker,** while 5 layers is only slightly better than 4, indicating limited gains beyond 4 layers. We therefore **choose 4 layers as the best trade-off between performance and computational cost.**
>
> [1] Relational Knowledge Distillation [CVPR 2019]
>
> [2] Comparative Knowledge Distillation [WACV 2025]

---

> > ### Author Rebuttal · Reviewer_bXmH · 2026-04-04
> >
> > Thank you for the detailed response and additional results, which addressed some of my concerns.
> >
> > - For W3, HieRD does not consistently have a better performance and only outperforms on two out of five subsets. For Q1, its margin over RKD on average performance is small (78.88 vs 78.43).
> > - Additionally, as Reviewer `jgvf` pointed out, the differences between methods are minor, possibly due to randomness in evaluation.
> >
> > Considering HieRD's higher computational cost (on both time and memory) and method complexity, its improvement over prior methods is not sufficiently significant. Thus, I maintain my original score.

---

> > > ### Author Response · Authors · 2026-04-04
> > >
> > > We thank the reviewer for the additional feedback and address the remaining concerns below.
> > >
> > > **1. On OOD generalization (W3).**
> > >
> > > We agree that OOD transfer is challenging and do not claim HieRD is best on every subset. However, the added OOD results still support **stronger aggregate robustness** (Link: **https://anonymous.4open.science/r/rebuttal-c9cc/exp_OOD_Evaluation.png**). HieRD achieves the **best overall OOD average (35.1)**, outperforming the strongest baselines **EM-KD (33.0)** and **CKD (32.8)** by **+2.1** and **+2.3**, respectively.
> > >
> > > At the subset level, HieRD remains consistently competitive: it is **top-1 on 2/5 OOD datasets, top-2 on 2/5**, and on the remaining one is still **very close to the best**. For example, HieRD achieves the best result on **ImageNet-A (32.7)** with a clear **+1.0** margin over the second-best, and on **Place365** it is only **0.4 below the best**. We therefore believe the OOD evidence supports our claim that HieRD improves robustness overall, even if not every individual subset is won.
> > >
> > > **2. On grounding performance and the magnitude of the gain (Q1).**
> > >
> > > We agree that the gain on grounding is more modest than on CLS/VQA. However, we would like to emphasize that HieRD remains consistently competitive across the grounding benchmarks, including the OOD subsets, and still achieves the best overall average (Link: **https://anonymous.4open.science/r/rebuttal-c9cc/exp_grounding.png**). HieRD achieves the **best overall grounding average (78.88)**, improving over the strongest baseline **RKD (78.43)** by **+0.45.**
> > >
> > > Importantly, this is not driven by one isolated subset: HieRD is **top-1 on 3/4 grounding datasets** and remains **very close to the best** on the remaining one, including the **OOD grounding** dataset. For instance, it is best on **MSCOCO, RefCOCO, and Visual7W-Pointing**, while on **RefCOCO-Matching** it is only **0.4** below the best. Thus, although the absolute gains are smaller here, the pattern is still stable: HieRD is consistently **best or near-best** across grounding benchmarks, including under distribution shift.
> > >
> > > **3. On robustness beyond single-run variability (also addressing the concern raised by Reviewer `jgvf`).**
> > >
> > > We would also like to clarify that our conclusions are not based on isolated single-run improvements. To address the concern that some differences might be attributable to randomness, we re-ran all experiments from the main Table 1 with three random seeds and report mean ± std over runs (Link: **https://anonymous.4open.science/r/rebuttal-c9cc/exp_multirun.png**).
> > >
> > > These results show that the gains remain consistent across both student backbones and both task groups.
> > > - For **FastVLM-0.5B**, HieRD achieves the **best average** on **CLS (67.11 ± 0.06)** and **VQA (52.76 ± 0.03)**. In **CLS**, it is **best on 4/5 datasets (ImageNet-1K, N24News, VOC2007, SUN397)** and improves over the **strongest baseline RKD (66.17 ± 0.14)** by **+0.94 points on average**. In **VQA**, it is best on **3/6 datasets (A-OKVQA, DocVQA, InfographicsVQA)**, second-best on **OK-VQA**, and still **improves over the strongest baseline EM-KD (51.56 ± 0.05)** by **+1.20 points on average**.
> > > - For **LLaVA-OneVision-0.5B**, the trend is even clearer: HieRD is **best on all CLS datasets and all VQA datasets**, with average **gains of +3.13 points** on **CLS (70.08 ± 0.05 vs. 66.95 ± 0.24, over CKD)** and **+1.87 points on VQA (42.25 ± 0.06 vs. 40.38 ± 0.13, over CKD).**
> > >
> > > Together with the added OOD, grounding results, multi-run results, we believe this directly addresses the concern—also raised by Reviewer jgvf—that the observed improvements may simply come from evaluation randomness. While some individual dataset-level margins are naturally small, the broader pattern is consistent: HieRD repeatedly achieves the strongest overall averages across multiple seeds, two student backbones, CLS, VQA, OOD, and grounding.
> > >
> > > **4. On computational cost versus benefit.**
> > >
> > > We agree that efficiency is important. However, the additional cost of HieRD is modest relative to the gains obtained. As shown in the efficiency table (Link: **https://anonymous.4open.science/r/rebuttal-c9cc/exp_computation_comparision.png**), HieRD requires **3.6 s/iter** and **63.5 GB VRAM**, compared with **3.3 s/iter for CKD/EMO**, and it even uses less memory than **EM-KD (63.5 GB vs. 64.8 GB)**. Since this overhead is incurred only during training, while the inference-time student remains unchanged, we believe this is a reasonable trade-off for the consistently stronger results across multi-run evaluation, OOD evaluation, and grounding.
> > >
> > > We hope these more detailed clarifications and additional results help address your remaining concerns.

---

### Decision · Program_Chairs · 2026-04-30

**Decision:**

Accept (regular)

**Comment:**

This paper presents a knowledge distillation approach for VLMs that adds structure to feature representations during training via clustering.  Reviewers had two notable concerns that were unresolved: 1) lack of generalization of this approach to multiple VLM families, 2) some reported gains may not be statistically significant.  For the first weakness, one could argue that the Qwen family is popular and, thus, even if the impact was limited to this style of model it would still be beneficial.  For the second weakness, the authors argue about the expense required to have multiple runs, but this is not convincing (i.e., sometimes it is expensive to do thorough research).  Further, while some reviewers note that the clustering + alignment is novel in this context, it closely resembles other VLM work [A], limiting its impact.  What is left is a paper that provides some contribution, although perhaps a bit limited.  That said, on the balance the AC finds that while further iterations would be helpful, but that the main idea of the work has been sufficiently presented and is different enough from prior work that some readers would likely find it interesting.  As such, the paper is recommended to be accepted if there is sufficient room in the program.


[A] From Coarse to Fine-grained Concept based Discrimination for Phrase Detection. CVPRW, 2023